# DriveRX: A Vision-Language Reasoning Model for Cross-Task Autonomous Driving

## Abstract

Effective autonomous driving hinges on robust reasoning across perception, pre-
diction, planning, and behavior. However, conventional end-to-end models fail to
generalize in complex scenarios due to the lack of structured reasoning. While
recent vision-language models (VLMs) have been applied to driving tasks, they
typically rely on isolated modules and static supervision, limiting their ability to
support multi-stage decision-making. We present **AutoDriveRL**, a unified training
framework that formulates autonomous driving as a structured reasoning process
over four core tasks. Each task is independently modeled as a vision-language QA
problem and optimized using task-specific reward models, enabling fine-grained
reinforcement signals at different reasoning stages. Within this framework, we train
**DriveRX**, a cross-task reasoning VLM designed for multi-stage decision-making.
DriveRX achieves strong performance on the public benchmark, outperforming
GPT-4o in behavior reasoning and demonstrating robustness under complex or
corrupted driving conditions. DriveRX serves as a high-level semantic reasoning
backbone, producing structured stage-wise reasoning chains that enhance decision
consistency. These outputs also provide high-quality supervisory signals for anno-
tation and downstream planning/control models. We will release the AutoDriveRL
framework and DriveRX to support future research.

## 1 Introduction

Autonomous driving systems operate in dynamic and uncertain traffic environments (Galceran et al.,
2015; Xu et al., 2014; Marcu et al., 2024), requiring robust capabilities in perception, prediction,
and decision-making (Grigorescu et al., 2020; Chen et al., 2015; Janai et al., 2020). However, in
complex scenarios such as occlusion or abnormal behaviors, conventional end-to-end models exhibit
limited generalization due to the lack of reasoning ability (Chen et al., 2024a; Xu et al., 2024b; Shao
et al., 2023). Moreover, EU regulations and academic research emphasize that autonomous driving
systems must be interpretable (Atakishiyev et al., 2023; 2024), highlighting the need for models with
structured reasoning and semantic understanding.

Vision-Language Models (VLMs) offer a promising way to meet this demand by combining perception
and reasoning. Recent progress has extended their applications to autonomous driving, including
object detection (Liu et al., 2023; Choudhary et al., 2024; Qian et al., 2024), trajectory prediction
(Mao et al., 2023; Nguyen et al., 2024; Tian et al., 2024), and planning (Jiang et al., 2025; Tian et al.,
2024). Prior work (Xu et al., 2024a; Tian et al., 2024; Wang et al., 2024) has demonstrated VLMs'
potential through language prompts and QA mechanisms, but most approaches primarily focus on
isolated task modules without a unified reasoning framework. This fragmentation hinders systematic
understanding and cross-task generalization (Zhao et al., 2025; Wang et al., 2024).

Building on these observations, existing VLM-based approaches to autonomous driving exhibit
several common limitations: **(1) absence of intermediate reasoning chains (Nie et al., 2024; Jiang
et al., 2025; Xu et al., 2024a); (2) isolation across perception, prediction, planning, and behavior
(Hu et al., 2023; Bansal et al., 2018; Casas et al., 2021); (3) reliance on static prompts and
annotations without adaptive feedback (Zhang et al., 2024; Cui et al., 2024; Duan et al., 2024).**

Meanwhile, Large Language Models (LLMs) have shown promising solutions to similar challenges
(OpenAI, 2024a; Team, 2024; Guo et al., 2025; OpenAI, 2024b). OpenAI's o1 (OpenAI, 2024b) lever-
ages Chain-of-Thought (CoT) prompting to generate intermediate reasoning steps, while DeepSeek's

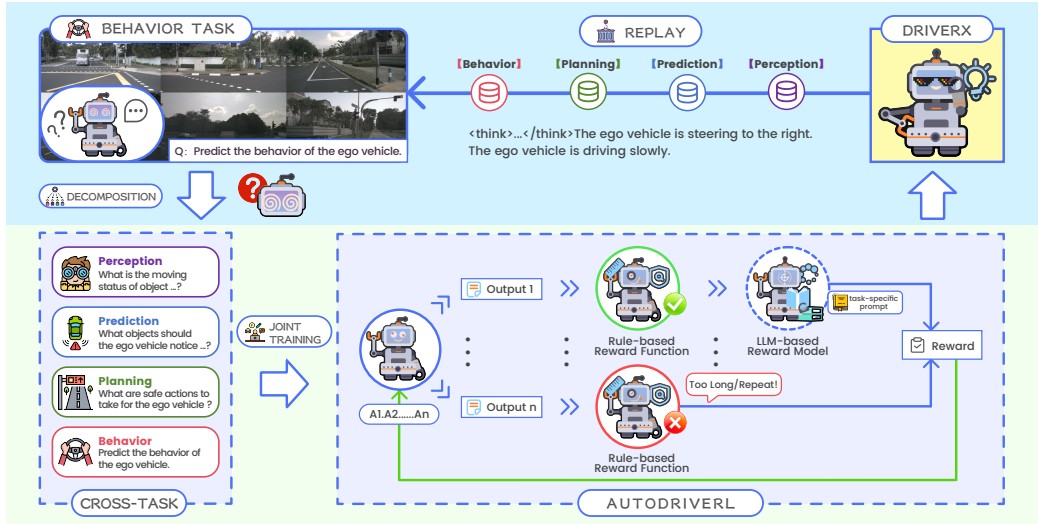

Figure 1: Overview of the **AutoDriveRL** framework. To tackle the challenge of cross-task reasoning in autonomous driving, AutoDriveRL decomposes complex scenarios into four core tasks—Perception, Prediction, Planning and Behavior—each formulated in a VQA style. These tasks form a structured reasoning chain and are jointly optimized via RL. During training, we design task-specific reward models for each task to provide fine-grained feedback. The resulting model, **DriveRX**, demonstrates strong generalization and robustness under challenging driving conditions.

R1-Zero (Guo et al., 2025) uses Reinforcement Learning (RL) to improve model behavior through dynamic feedback—without relying on supervised fine-tuning. These approaches address key limitations in current VLMs by enabling structured reasoning, unifying task semantics, and supporting feedback-driven optimization, motivating their extension to autonomous driving.

Several multimodal studies have explored integrating reasoning mechanisms to enhance model capabilities. LlamaV-o1 (Thawakar et al., 2025) combines Monte Carlo Tree Search with CoT prompting to improve logical consistency in Visual Question Answering (VQA); VLM-R1 (Shen et al., 2025) and MM-Eureka (Meng et al., 2025) apply RL and rule-based supervision to extend reasoning into science domains. While effective in their domains, these approaches are not designed for the decision-critical demands of autonomous driving. Long-form reasoning chains, common in CoT-based methods, are often exhibit excessive length and latency for driving scenarios that require concise, executable outputs. AlphaDrive (Jiang et al., 2025) explores the application of RL in autonomous driving, but only within the planning task, and does not address multi-task coordination.

This motivates formulating autonomous driving as a structured reasoning process across four stages—perception, prediction, planning, and behavior—and developing a unified vision-language framework for coherent cross-task semantic understanding.

To this end, we introduce AutoDriveRL, which jointly optimizes a unified VLM through task-specific RL signals that enhance each reasoning stage and improve cross-task consistency and robustness.

Building on this four-stage formulation, AutoDriveRL models each stage as an independent VQA problem organized into a sequential reasoning chain. The framework incorporates task-specific reward models that deliver fine-grained, stage-level reinforcement signals, enabling process-level feedback across reasoning steps. This structured supervision supports interpretable intermediate reasoning, facilitates cross-task optimization, and substantially improves the stability and overall performance of autonomous driving decision making.

Within this framework, we train a unified cross-task vision-language model, DriveRX, where "RX" emphasizing reasoning and cross-task generalization in autonomous driving. DriveRX consistently outperforms existing models across perception, prediction, planning, and behavior tasks, in both in-distribution and out-of-distribution (OOD) benchmark settings. On the public benchmark DriveBench (Xie et al., 2025), DriveRX achieves a score of 62.02 in behavior, surpassing GPT-4o's (OpenAI, 2024a) 55.04 by 6.98 points. It produces stable outputs under corruption-based perturbations (e.g.,

rain, occlusion, lighting variation), demonstrating strong robustness to visual degradation. DriveRX obtains 55.24 on DriveBench-Corruption, outperforming GPT-4o (53.97), and further shows strong performance on DriveLM-Hard, which focuses on semantically challenging, long-tail scenarios. Together, these results confirm that DriveRX provides stable and interpretable reasoning chains for high-level semantic understanding and consistent cross-task decision making.

Based on DriveRX, we investigate two downstream applications that leverage its high-level reasoning outputs as structured supervisory signals. First, we extend the vocabulary of DriveRX to build the DriveRX-Agent for trajectory prediction. In open-loop testing on nuScenes, DriveRX-Agent surpasses the baseline DriveLM-Agent, indicating that the reasoning chains of DriveRX offer more reliable guidance in complex, dynamic environments. Second, we distill DriveRX's stage-wise reasoning data into a compact Vision-Language-Action (VLA) model. The reasoning-enhanced supervision substantially improves action prediction quality, enabling the distilled model to achieve competitive closed-loop performance on Bench2Drive. Together, these results validate that DriveRX's structured reasoning serves as effective teacher signals or reward feedback for downstream planning and control models, benefiting both trajectory prediction and action generation.

Our main contributions are as follows:

- **Propose AutoDriveRL**: We propose AutoDriveRL, a unified RL framework that designs task-specific reward models for four core autonomous driving tasks—perception, prediction, planning, and behavior—and jointly optimizes them to enable structured, interpretable reasoning across tasks.

- **Develop DriveRX**: We develop DriveRX, a cross-task VLM that demonstrates strong performance across multiple driving tasks and complex traffic scenarios, with strong generalization and robustness, and serving as a reliable backbone for high-level semantic understanding and reasoning-based analysis in autonomous driving.

- **Explore Applications of VLM**: We further explore two downstream applications that leverage DriveRX's structured reasoning outputs as supervisory signals: (i) DriveRX-Agent, which extends the vocabulary for trajectory prediction and achieves competitive open-loop performance; and (ii) DriveRX-VLA, a compact VLA model distilled from DriveRX's reasoning data, showing strong closed-loop results on Bench2Drive. These applications demonstrate the versatility and effectiveness of reasoning-enhanced VLMs for downstream planning and control.

## 2 RELATED WORK

### 2.1 VISION-LANGUAGE MODELS FOR AUTONOMOUS DRIVING

Recent works have explored using vision-language models (VLMs) to enhance perception and reasoning in autonomous driving. DriveLM (Sima et al., 2024) formulates multiple driving tasks as structured graph-based QA, constructing a multi-task dataset. Dolphins (Ma et al., 2024) focuses on dialogue-based reasoning with historical control and counterfactuals. DriveVLM (Tian et al., 2024) proposes a three-stage language-driven planning pipeline with templated prompts and task-specific instructions. These methods largely rely on supervised fine-tuning (SFT) over static human-annotated data for task-specific outputs. AlphaDrive (Jiang et al., 2025) introduces reinforcement learning (RL) but is limited to the planning stage, lacking support for end-to-end reasoning and broader task-level optimization. In contrast, we adopt an RL framework that decomposes driving into four semantically distinct subtasks, each equipped with task-specific rewards to evaluate response quality and guide policy learning. This enables the model to acquire task-aligned reasoning strategies at every stage without static supervision, improving robustness and consistency in complex real-world scenarios.

### 2.2 VISION-LANGUAGE REASONING MODELS

Recent years have seen notable progress in VLMs for visual reasoning. Representative works include R1-Onevision (Yang et al., 2025b), which enhances visual-textual reasoning via structured intermediate representations and employs reinforcement learning to optimize reasoning trajectories; MM-Eureka (Meng et al., 2025), which introduces rule-based "aha" mechanisms for iterative refinement; LlamaV-o1 (Thawakar et al., 2025), which adopts step-by-step prompting and achieves strong performance; and VisuoThink (Wang et al., 2025), which leverages multimodal tree search to emulate human-like slow thinking, achieving strong performance in geometry and spatial reasoning tasks. These models are primarily developed for general-purpose vision-language reasoning, typically under static image and single-turn QA settings. They seldom address complex task structures or action-oriented reasoning, making them less applicable to domains like autonomous driving that require long-horizon, context-dependent reasoning across multiple subtasks.

# 3 AUTODRIVERL

To enhance the reasoning capability and generalization of VLMs in complex driving scenarios, we propose **AutoDriveRL**, a unified training framework. AutoDriveRL decomposes the autonomous driving task into four core subtasks and adopts a consistent question-answering structure across all tasks. Each sub-task is equipped with a task-specific reward function, providing targeted feedback to guide the model toward learning stage-specific reasoning strategies. These components are jointly trained using reinforcement learning to develop a multi-task vision-language model, **DriveRX**. In this section, we detail the approach across four dimensions: task formulation, data construction, training framework, and reward design. An overview of the overall pipeline is illustrated in Figure 1.

## 3.1 TASK FORMULATION

Autonomous driving involves perceiving traffic elements, anticipating agent behaviors, and making timely planning and behavioral decisions in dynamic environments. Following prior work (Sima et al., 2024; Xie et al., 2025), we decompose this process into four sub-tasks—perception, prediction, planning, and behavior—that span the full reasoning chain from low-level perception to high-level decision-making.

- **Perception:** Identify key elements in the scene, such as vehicles, pedestrians, and traffic signs.

- **Prediction:** Anticipate the behavior of dynamic agents.

- **Planning:** Determine near-term path or directional actions.

- **Behavior:** Conduct behavior prediction and strategy selection. This task provides a discrete summary of the ego vehicle's final driving decision (e.g., "slowly go straight").

Each task is formulated as VQA problems, where the input consists of images and a task-specific natural language query, and the output is a grounded, interpretable response. The tasks share a unified input-output structure to facilitate consistent language-based reasoning, and are shuffled and jointly trained using reinforcement learning to encourage generalizable reasoning across task boundaries.

Although trained jointly with shared parameters, the tasks form an implicit reasoning chain in semantic space. We do not explicitly condition later tasks on earlier outputs; instead, all tasks attend to a shared image and question, allowing semantic coupling to emerge through language.

In addition, the Behavior stage provides a lightweight and interpretable interface between high-level planning intents and downstream action or trajectory generation, yielding a clearer semantic hierarchy and offering stable supervision for subsequent VLA learning.

## 3.2 DATA CONSTRUCTION

To ensure stable RL training and challenging evaluation, we construct two subsets from the DriveLM dataset: **AutoDriveRL-Train** for multi-task training and **DriveLM-Hard** for robustness evaluation.

We employ a unified scoring and filtering pipeline to control task difficulty, increase semantic diversity, and balance action distributions. Low-quality samples (e.g., with annotation errors or duplicate options) are first removed. For each visual question, we collect response scores $s_{ijkm}$ by sampling $K$ outputs from each of $M$ VLMs. These scores are computed using task-specific reward models (see prompt in Section M.1), guided by structured prompts to ensure consistency and reproducibility without human labeling. Alignment with human judgments is shown in Appendix I.

Each image frame typically contains multiple questions. We define a frame-level score $D_i$ as:

$$D_i = \frac{1}{T_i} \sum_{j=1}^{T_i} \left( \frac{1}{M} \sum_{k=1}^{M} \left( \frac{1}{K} \sum_{m=1}^{K} s_{ijkm} \right) \right) \tag{1}$$

where $T_i$ is the number of questions in frame $i$, $M$ is the number of base models used for scoring, and $K$ is the number of sampled responses per model per question. Each score $s_{ijkm} \in [0, 100]$ reflects the task-specific quality of a sampled response.

Based on $D_i$, we follow prior work (Guo et al., 2025; Zeng et al., 2025) on RL data filtering strategies and exclude both overly easy and overly hard samples from training. Specifically, we select mid-range frames to improve training stability by avoiding trivial examples and high-variance outliers. Low-scoring frames—typically involving occlusion, poor lighting, or rare behaviors—are used exclusively for evaluation to assess generalization under challenging conditions.

To mitigate action imbalance, we oversample rare actions, improving coverage of low-frequency behaviors. Final distributions are shown in Figure 6. These process yields a high-quality, difficulty-controlled, and semantically diverse dataset that supports robust multi-task RL and evaluation.

### 3.3 TRAINING FRAMEWORK

Autonomous driving comprises four semantically coupled sub-tasks—perception, prediction, planning, and behavior decision-making—each requiring intermediate reasoning that is hard to supervise directly. Instead of conventional supervised learning, we adopt a reinforcement-learning paradigm in which a single policy is trained across all tasks and updated from reward feedback.

Many tasks, especially planning and behavior prediction, admit multiple plausible outputs (*one-to-many* mapping). To cope with the resulting high-variance rewards, we optimize the policy with Group Relative Policy Optimization (GRPO) (Shao et al., 2024). For every query $q$, the current policy $\pi_\theta$ samples $G$ candidate responses $\{o_1, \ldots, o_G\}$. The GRPO objective is

$$\mathcal{J}_{\text{GRPO}}(\theta) = \mathbb{E}_{q \sim \mathcal{D}} \left[ \frac{1}{G} \sum_{i=1}^{G} \frac{1}{|o_i|} \sum_{t=1}^{|o_i|} \min\left( r_{i,t} \hat{A}_{i,t}, \text{clip}(r_{i,t}, 1 - \epsilon, 1 + \epsilon) \hat{A}_{i,t} \right) \right] - \beta \, \text{KL}\left( \pi_\theta \parallel \pi_{\text{ref}} \right),$$

(2)

where $r_{i,t} = \pi_\theta(o_{i,t} \mid q, o_{i,<t}) / \pi_{\theta_{\text{old}}}(o_{i,t} \mid q, o_{i,<t})$ is the token-level policy ratio, $t$ denotes the token position and $\hat{A}_{i,t}$ is the intra-group advantage defined in Section 3.4.

During training, samples from all four tasks are mixed, and the shared parameters are updated jointly. The framework relies solely on rule-based and model-based reward signals, requiring no extra human annotation and thus scaling easily to new tasks and datasets.

### 3.4 REWARD MODEL DESIGN

Autonomous driving demands low latency, clear expression, and accurate actions. To encourage concise and reliable outputs while avoiding verbosity or repetition, we design a two-stage reward mechanism that automatically evaluates response quality and guides GRPO training.

**(i) Rule-based Reward Function.** Outputs exceeding a predefined length or exhibiting repetitive sentence patterns are penalized with low rewards.

**(ii) LLM-based Reward Model.** For remaining candidates, we employ an open-source language model to assign a task-aware quality score $R_i \in [0, 100]$. Crucially, as detailed in Appendix M.1, distinct prompts and evaluation rubrics are tailored to each of the four sub-tasks to address their specific requirements—ranging from coordinate precision in Perception to safety logic in Planning. Despite these domain differences, all tasks utilize this unified scalar range, ensuring that reinforcement signals remain consistent and directly comparable across the heterogeneous tasks. This standardization prevents optimization instability caused by varying score magnitudes and facilitates effective joint training. The normalized group-level advantage is computed as:

$$\hat{A}i = \frac{R_i - \mu_R}{\sigma_R}, \quad \hat{A}_{i,t} = \hat{A}_i, ; \forall t,$$

(3)

where $\mu_R$ and $\sigma_R$ are computed over the $G$ responses sampled for the same input. This scalar advantage is shared across tokens and used in the GRPO update in Eq. 2.

## 4 EXPERIMENT

This section introduces the training setup of **DriveRX** with **AutoDriveRL** and evaluates the resulting model on public out-of-distribution (OOD) benchmarks.

## 4.1 EXPERIMENT SETTINGS

**Base Model:** We adopt **Align-DS-V**(PKU-Alignment, 2025) as our base model, which extends DeepSeek-R1-Distill-LLaMA-8B(Guo et al., 2025) with a CLIP-ViT-L/14-336 visual encoder (Radford et al., 2021), trained using the Align-Anything framework (Ji et al., 2024), and exhibits strong multi-modal reasoning performance.

**Evaluated Models:** We evaluate a diverse set of models, including the commercial GPT-4o (2024-08-06) (OpenAI, 2024a), the general open-source model LLaVA-1.5 7B (Liu et al., 2024a), LLaVA-NeXT 7B (Liu et al., 2024b), InternVL3 8B/78B (Zhu et al., 2025), Qwen2.5-VL 7B/72B (Bai et al., 2025), the reasoning MM-Eureka 7B (Meng et al., 2025), R1-Onevision 7B (Yang et al., 2025b), and the domain-specific driving models DriveLM (Sima et al., 2024) and Dolphins (Ma et al., 2024).

**Training Dataset:** We use the DriveLM (Sima et al., 2024) dataset, which is constructed from nuScenes (Caesar et al., 2020) and OpenLane-V2 (Wang et al., 2023) through a structured annotation process. Following the AutoDriveRL data selection strategy, we compute frame-level scores $D_i$ using Align-DS-V and retain samples within the range $[25, 45]$. To ensure a non-overlapping evaluation, all samples appearing in DriveBench (Sima et al., 2024) are explicitly excluded. From the remaining subset, we randomly sample 6K instances for training. See Appendix C for the dataset composition.

**Reward:** We first apply a rule-based filter that assigns a reward of 0 to responses exceeding 4K tokens or exhibiting substantial repetition. The remaining responses are then evaluated using task-specific prompts by Qwen2.5-72B-Instruct (Team, 2024) (see Appendix M.1). The consistency between human judgments and model scores, as well as the scoring latency, is analyzed in Appendix I and Appendix O.3, respectively.

**Evaluation:** We evaluate on DriveBench (Sima et al., 2024), which spans four driving tasks and consists of a *clean* set with re-sampled questions and a *corruption* set featuring 15 types of visual degradation (see Appendix N) for robustness testing. We additionally evaluate on **DriveLM-Hard**, a challenging subset selected using the AutoDriveRL data selection strategy, where frame-level scores $D_i$ (defined in Eq. 1) fall within the range $[10, 31]$. The $D_i$ scores are computed by averaging model scores from Align-DS-V, Qwen2-VL-7B, and Qwen2-VL-72B.

**Metrics:** We follow DriveBench evaluation prompts and use GPT-3.5-Turbo as the judge model, with GPT score as the evaluation metric. Following prior work (Sima et al., 2024; Tran et al., 2019; Anderson et al., 2016; Akter et al., 2022; Evtikhiev et al., 2023), we discard BLEU (Papineni et al., 2002) and ROUGE (Lin, 2004), as they were designed for translation and are not suitable for evaluation in autonomous driving. For more experimental setup and metrics details in Appendix O.

## 4.2 MAIN RESULT

**DriveRX shows strong overall performance and achieves the best result in behavior tasks:** As shown in Table 1, DriveRX demonstrates competitive performance across all four tasks. On the clean test sets of **Perception**, **Prediction**, and **Planning**, it performs closely to GPT-4o and the larger 70B-scale models, despite its smaller size. Notably, it achieves the highest score ($\mathbf{62.02}$) on the critical **Behavior** task, surpassing all baselines including the closed-source GPT-4o (55.04), the open-source Qwen2.5-VL 72B (55.57), the reasoning model MM-Eureka (50.50), and the domain-specific DriveLM (42.78). These results highlight the effectiveness of the AutoDriveRL framework, as improvements in earlier tasks lead to significant gains in the final behavior task.

**DriveRX maintains robust performance under visual corruptions:** In Table 1, DriveRX exhibits consistently strong performance across all tasks under the corruption setting, demonstrating resilience to various types of visual degradation. Notably, it achieves the highest score of $\mathbf{55.24}$ on the **Behavior** task, outperforming all other models. This result highlights the robustness of DriveRX in handling distributional shifts, and further demonstrates the benefit of RL: the model receives step-wise feedback during training, enabling reliable decision-making even under visual corruptions.

**DriveRX achieves stable performance in complex traffic conditions:** As shown in Table 2, DriveRX performs strongly on DriveLM-Hard, a challenging test split we construct by selecting high-difficulty frames from the DriveLM dataset. This benchmark features congested traffic, diverse agents, and denser interactions across the same four tasks (see Appendix L). These results demonstrate that DriveRX maintains reliable performance in complex and high-uncertainty scenarios, confirming

Table 1: **Evaluations of VLMs across different driving tasks on DriveBench** (perception, prediction, planning, and behavior). "Clean" represents clean image inputs. "Corr." represents corruption image inputs, averaged across fifteen corruptions. The evaluations are based on GPT scores (↑), where we tailored detailed rubrics for each task and question type. We highlight the best-performing open-source model for each task, and use **bold** to mark the second-best performance.

| Model | Size | Type | Perception | | Prediction | | Planning | | Behavior | |
|---|---|---|---|---|---|---|---|---|---|---|
| | | | Clean | Corr. | Clean | Corr. | Clean | Corr. | Clean | Corr. |
| GPT-4o (OpenAI, 2024a) | - | Commercial | 41.51 | 45.39 | 52.28 | 50.92 | 84.82 | 84.43 | 55.04 | 53.97 |
| LLaVA-1.5 (Liu et al., 2024a) | 7 B | Open | 23.22 | 22.95 | 22.02 | 17.54 | 29.15 | 31.51 | 13.60 | 13.62 |
| LLaVA-NeXT (Liu et al., 2024b) | 7 B | Open | 24.15 | 19.62 | 35.07 | 35.89 | 45.27 | 44.36 | 48.16 | 39.44 |
| Qwen2.5-VL (Team, 2024) | 7 B | Open | 31.92 | 27.16 | 42.26 | 47.13 | 55.70 | 57.97 | 47.40 | 42.82 |
| InternVL3 (Zhu et al., 2025) | 8 B | Open | 33.21 | **33.87** | 41.21 | 38.93 | 70.97 | 56.71 | 51.47 | 50.32 |
| Qwen2.5-VL (Team, 2024) | 72 B | Open | **37.56** | 32.25 | 54.89 | 50.47 | 77.18 | 74.01 | **55.57** | 51.65 |
| InternVL3 (Zhu et al., 2025) | 78 B | Open | 39.44 | 35.62 | 50.93 | **49.71** | 82.24 | 78.41 | 50.73 | **52.03** |
| MM-Eureka (Meng et al., 2025) | 7 B | Reasoning | 34.44 | 28.92 | 40.31 | 41.72 | 65.38 | 60.18 | 49.40 | 47.59 |
| R1-Onevision (Yang et al., 2025b) | 7 B | Reasoning | 28.64 | 25.40 | 51.22 | 46.91 | 55.12 | 55.54 | 41.57 | 39.88 |
| DriveLM (Sima et al., 2024) | 7 B | Specialist | 16.85 | 16.00 | 44.33 | 39.71 | 68.71 | 67.60 | 42.78 | 40.37 |
| Dolphins (Ma et al., 2024) | 7 B | Specialist | 9.59 | 10.84 | 32.66 | 29.88 | 52.91 | 53.77 | 8.81 | 8.25 |
| Align-DS-V (PKU-Alignment, 2025) | 8 B | Reasoning | 31.41 | 27.78 | 41.51 | 37.86 | 51.16 | 57.66 | 50.53 | 40.98 |
| DriveRX | 8 B | Reasoning | 32.88 | 29.92 | **52.20** | 48.29 | **78.63** | **75.53** | 62.02 | 55.24 |

Table 2: **Evaluations of VLMs across different driving tasks on DriveLM-Hard.** Each column shows the GPT score (↑) for the clean version of each task. We highlight the best-performing open-source model for each task, and use **bold** to mark the second-best performance.

| Method | Size | Type | Perception | Prediction | Planning | Behavior |
|---|---|---|---|---|---|---|
| GPT-4o (OpenAI, 2024a) | - | Commercial | 36.12 | 65.90 | 51.42 | 42.00 |
| Qwen2.5-VL (Bai et al., 2025) | 7B | Open | 26.10 | 60.13 | 32.41 | 19.86 |
| InternVL3 (Zhu et al., 2025) | 8B | Open | 31.29 | 38.95 | 31.78 | 30.48 |
| Qwen2.5-VL (Bai et al., 2025) | 72B | Open | 32.49 | 58.63 | **50.55** | 40.02 |
| InternVL3 (Zhu et al., 2025) | 78B | Open | **32.98** | 58.67 | 45.82 | 35.69 |
| MM-Eureka (Meng et al., 2025) | 7B | Reasoning | 29.09 | **67.41** | 29.67 | 23.21 |
| R1-Onevision (Yang et al., 2025b) | 7B | Reasoning | 25.51 | 46.60 | 34.45 | 29.03 |
| DriveLM (Sima et al., 2024) | 7B | Specialist | 28.78 | 19.93 | 22.60 | 30.75 |
| Align-DS-V (PKU-Alignment, 2025) | 8B | Reasoning | 29.45 | 52.90 | 29.15 | 32.48 |
| DriveRX | 8B | Reasoning | 34.83 | 71.84 | 51.42 | **36.82** |

its robustness and generalization. Such capabilities are critical for safety-critical autonomous driving applications that demand consistent reasoning under diverse and dynamic conditions.

Overall, our results on DriveBench and DriveLM-Hard demonstrate that DriveRX consistently produces *stable and interpretable stage-wise reasoning* across challenging, corrupted, and long-tail scenarios. These findings confirm that DriveRX is well suited as a **high-level semantic reasoning backbone** for autonomous driving: its structured Perception→Prediction→Planning→Behavior chain provides reliable cross-task semantics and intention-level understanding.

## 5 TRAINING ANALYSIS

We analyze the training dynamics of **DriveRX** under AutoDriveRL to understand cross-task optimization. In Figure 5(a), the overall training remains stable, with steadily increasing rewards. This suggests that our task-specific reward models are effective in guiding learning across heterogeneous tasks. In Figure 5(b), the reward curves exhibit a clear staged trend: **Perception** and **Prediction** improve rapidly in early training and quickly saturate, while **Planning** and **Behavior** progress more slowly and gradually. This pattern reflects the structural hierarchy of the tasks—lower-level tasks are relatively simple, focusing on visual understanding and pattern recognition, whereas higher-level tasks require multi-step reasoning and rely on semantic representations generated by preceding tasks.

We also track the average response length during training (Figure 5(c)). In the early stages, response length increases, indicating that the model explores more expressive outputs. As training progresses and reward signals stabilize, responses gradually become shorter and more concise. Figure 5(d) shows the ratio of clipped responses exceeding the maximum length threshold, which closely follows

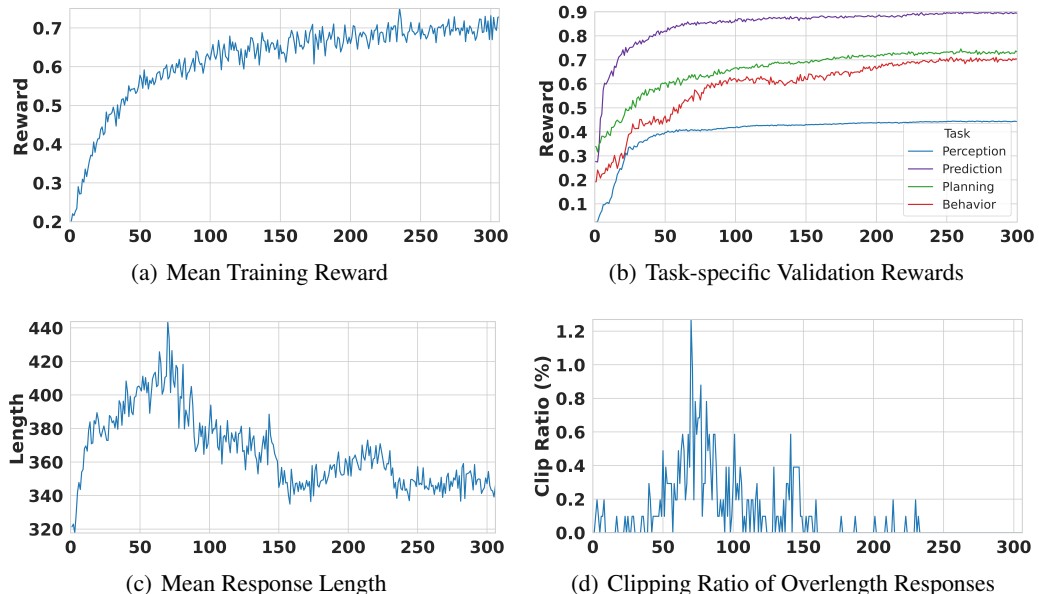

Figure 2: Training curves of DriveRX over reinforcement learning steps.

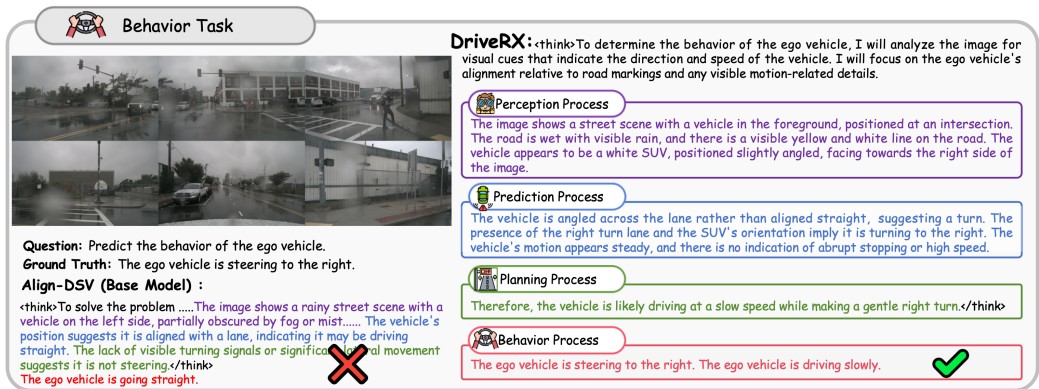

Figure 3: Behavior task example comparing Align-DSV and DriveRX. The goal is to determine the ego vehicle's behavior at an intersection. Align-DSV incorrectly outputs "going straight", while DriveRX correctly concludes "steering to the right" by performing structured reasoning. The behavior task is decomposed into four subtasks: Perception, Prediction, Planning, and final Behavior decision, allowing DriveRX to make more accurate and interpretable decisions.

the trend of response length—higher lengths lead to more clipping, while this ratio decreases as the model converges. These results confirm the effectiveness of our length-penalizing reward design, which encourages the model to reduce redundancy while maintaining output quality.

## 5.1 CASE STUDY

To better illustrate the advantages of our framework in terms of reasoning quality and interpretability, we present a representative example from the DriveLM-Hard set, featuring a challenging intersection scenario under rainy weather and blurred camera conditions. The task is to determine the correct behavior of the ego vehicle based solely on the visual input. We compare the outputs of the base model Align-DS-V and our RL-optimized model DriveRX.

As shown in Figure 3, DriveRX decomposes the behavior reasoning process into four semantically distinct subtasks: Perception, Prediction, Planning, and final Behavior, and answers them sequentially. This structured reasoning not only improves interpretability of the final decision, but also enables in-depth analysis of the intermediate steps.

**Perception:** Align-DS-V mainly provides scene-level descriptions (e.g., *"a rainy street scene with a vehicle on the left"*), while DriveRX focuses on driving-critical cues like the *wet road surface* and *vehicle pose*.

**Prediction:** Align-DS-V overlooks subtle orientation bias and mispredicts the vehicle's intent as going straight. In contrast, DriveRX correctly infers a right turn and estimates motion status (e.g., speed, steering).

**Planning:** Relying solely on explicit visual evidence (e.g., "no turn signal"), Align-DS-V struggles under incomplete cues. DriveRX integrates implicit context with earlier outputs, enabling more reliable planning in degraded environments.

**Behavior:** Due to its flawed reasoning chain, Align-DS-V makes an incorrect "go straight" decision, whereas DriveRX outputs the correct and interpretable action to *steer right*.

## 6 APPLICATIONS

We explore two applications where DriveRX serves as a high-level reasoning backbone that provides semantic supervision for downstream planning/control models: trajectory prediction and action generation. These applications illustrate how a reasoning-enhanced VLM can support cross-task semantic understanding and intention reasoning, and how its structured outputs can effectively guide downstream models in complex driving scenarios. Additional details are provided in Appendix H.

**DriveRX-Agent.** Following prior work (Zitkovich et al., 2023; Sima et al., 2024), we extend the vocabulary of DriveRX by discretizing the trajectory coordinates (the waypoints of the ego vehicle) into 512 bins, empirically determined based on the statistics of the training set trajectories. We then redefine the tokens in the DriveRX language tokenizer, creating distinct tokens for each bin, and fine-tune the VLM using this redefined vocabulary. This enables DriveRX-Agent to directly generate the cor-

Table 3: Open-loop testing on nuScenes.

| Methods | ADE↓ | Col.↓ |
|---|---|---|
| Command Mean | 4.57 | 5.71 |
| BLIP-RT-2 | 2.63 | 2.77 |
| DriveLM-Agent | 1.39 | 1.67 |
| DriveRX-Agent | 1.53 | 1.12 |

responding vehicle trajectory. We conduct open-loop evaluation on the nuScenes dataset, computing the ADE and Collision Rate (Col.) by averaging the values at the 1s, 2s, and 3s time horizons, following the evaluation settings in DriveLM (Sima et al., 2024), and the results in Table 3 show that DriveRX-Agent outperforms the baseline DriveLM-Agent, highlighting how improved reasoning in VLMs can directly translate into better trajectory prediction.

**DriveRX-VLA.** We further distill reasoning data from DriveRX and use supervised fine-tuning to train a smaller Vision-Language-Action(VLA) model based on the Qwen2.5-VL-3B (Bai et al., 2025). The prompt set is derived from the DriveLM-CARLA (Sima et al., 2024). Continuous vehicle trajectories are discretized into a series of physical action tokens using a K-means clustering approach, forming an action codebook with 2048 discrete tokens representing short-term spatial position changes and heading movements (steer, throttle, brake). These tokens are integrated into the VLM vocabulary, allowing the model to directly predict control actions during inference. Trained on reasoning data from DriveRX, the resulting model is capable of generating appropriate actions even in complex driving scenarios. Closed-loop evaluations on the Bench2Drive (Jia et al., 2024) benchmark (see Table 4, the planning frequency is set to 2 Hz) show that DriveRX-VLA achieves competitive performance. In the evaluation, While DriveRX-VLA slightly lags behind the top-performing model, it nonetheless achieves competitive results using significantly less training data compared to Orion, thereby demonstrating the efficacy of reasoning-based distillation.

Overall, DriveRX provides stable and interpretable reasoning across four tasks, enabling it to serve as an effective high-level semantic backbone for downstream driving tasks. Its structured outputs offer high-quality supervisory signals—both as reasoning traces and reward feedback—that enhance trajectory prediction and action generation when distilled into smaller models. The results of DriveRX-Agent and DriveRX-VLA confirm the feasibility of leveraging a unified reasoning-enhanced VLM to improve downstream planning and control models.

## 7 CONCLUSION

In this work, we proposed **AutoDriveRL**, a RL framework that decomposes autonomous driving into four interpretable reasoning subtasks—perception, prediction, planning, and behavior—each guided

Table 4: Closed-loop Results on the Bench2Drive.

| Methods | Driving Score ↑ | Success Rate (%) ↑ | Efficiency ↑ | Comfortness ↑ |
|---|---|---|---|---|
| TCP-traj (Wu et al., 2022) | 59.90 | 30.00 | 76.54 | 18.08 |
| AD-MLP (Zhai et al., 2023) | 18.05 | 0.00 | 48.45 | 22.63 |
| UniAD-Base (Hu et al., 2023) | 45.81 | 16.36 | 129.21 | 43.58 |
| VAD (Jiang et al., 2023) | 42.35 | 15.00 | 157.94 | 46.01 |
| DriveTransformer-Large (Jia et al., 2025) | 64.22 | 33.08 | 70.22 | 16.01 |
| Orion (Fu et al., 2025) | 77.74 | 54.62 | 151.48 | 17.38 |
| DriveRX-VLA | 72.36 | 45.23 | 138.47 | 17.36 |

by a task-specific reward model. This design enables structured reasoning and improves decision interpretability. Based on this framework, we developed **DriveRX**, a reasoning-enhanced VLM that performs robust scene analysis across core driving tasks, achieving strong results and efficient inference under challenging conditions. Furthermore, we explored the downstream applications of reasoning-enhanced VLMs by implementing **DriveRX-Agent** for trajectory generation and **DriveRX-VLA** for action prediction. Both models achieve competitive performance in open- and closed-loop evaluations, demonstrating the feasibility and potential of a unified VLM backbone for supporting both high-level decision-making and low-level control.

## 8 REPRODUCIBILITY STATEMENT

To ensure the reproducibility of our work, we will publicly release all relevant resources, including training and inference scripts, as well as the processed DriveLM-Hard dataset used in our experiments. The code will be made available via a link to a public repository. Additionally, we provide detailed descriptions of the data preprocessing steps, including the custom data filtering strategy applied to the DriveLM-Hard dataset, which is described in the supplementary materials. The experimental setup, including hardware, software versions, and model configurations, will also be documented in the appendix to facilitate accurate replication. We encourage other researchers to utilize these resources to reproduce and build upon our results.

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

# Appendix

CONTENTS

## A    LIMITATIONS AND BROADER IMPACT

In this paper, we introduce AutoDriveRL, a unified training framework for vision-language models in autonomous driving that enhances reasoning and generalization across four core driving tasks; however, the approach still faces practical limitations.

**Reward Function Design.** AutoDriveRL relies on manually designed, task-specific reward functions, which may limit its scalability to novel tasks or sensor modalities. Developing more generalizable or learnable reward signals remains an important direction for future research.

**Efficiency Constraints.** Inheriting the drawbacks of large language models (LLMs), our framework suffers from relatively long inference latency. Although our 8B-size model achieves performance comparable to or even surpassing 72B-scale models (see Table 1), the throughput remains a bottleneck, similar to prior works such as DriveLM (8.5 tokens/s in DriveLM-Agent). Unlike DriveLM, our approach does not require multi-round predictions, which leads to faster inference in practice. Moreover, recent advances in model optimization have improved LLM inference speed by orders of magnitude (Yang et al., 2025a; Bai et al., 2025), and we expect this trend to further alleviate the issue. We also experimented with a lightweight 3B VLA model, which significantly improves inference efficiency while maintaining competitive performance. In particular, the inference speed of Qwen2.5-VL-3B is approximately 2–3× faster than that of 8B models.

**Applications and Scope.** This section presents two application pathways of our framework — **DriveRX-Agent** for trajectory prediction and **DriveRX-VLA** for action-level control. These results show that injecting reasoning capabilities into a vision-language model (VLM) enables DriveRX to function as a high-level semantic reasoning backbone that provides structured cross-task understanding and intention reasoning for downstream planning and control models. Under the current experimental setup, both models achieve *competitive* performance in open- and closed-loop evaluations, though they still fall slightly short of the state-of-the-art (SOTA) on some metrics.

However, achieving the best low-level driving performance is *not* the primary goal of this work. Instead, our aim is to *verify the effectiveness and applicability of* reasoning-enhanced VLMs as a unified backbone for semantic perception, prediction, planning, and behavior abstraction in autonomous driving. While the structured reasoning outputs of DriveRX can benefit downstream planners and controllers, such extensions go beyond the scope of this paper and are left for future research. To this end, we adopt a simple and reproducible methodology, focusing on whether reasoning augmentation reliably improves high-level planning and semantic decision-making. Under this scope, DriveRX-VLA achieves competitive closed-loop results with limited data and minimal tuning, supporting our claim that a reasoning-based VLM can serve as a unified backbone for autonomous driving.

**Reasoning Failure Cases.** We observe that most failure cases arise in scenes where the Perception stage becomes unreliable due to severe occlusion or missing visual evidence. Since Perception initializes the entire reasoning chain, inaccuracies at this stage naturally propagate to subsequent Prediction, Planning, and Behavior stages, leading to inconsistencies across reasoning outputs. In addition, in a small number of low-confidence scenarios, DriveRX adopts a more conservative reasoning strategy, re-checking ambiguous cues and occasionally generating longer reasoning traces, which slightly increases the inference time. These behaviors occur only under highly uncertain visual conditions and do not affect overall performance or latency in normal cases.

Broader Impact. By promoting interpretable and modular reasoning, AutoDriveRL has the potential to advance the development of safer and more transparent autonomous systems. However, improper deployment of partially trained models or misaligned reward signals could lead to unsafe behavior. We encourage responsible use and further validation in realistic driving scenarios. Moreover, we plan to release **DriveRX** to the research community, hoping it can serve as a valuable source for distilling high-quality reasoning data and further accelerating progress in autonomous driving research. Importantly, the goal of this work is not to report improvements in real-time low-level driving performance, but to demonstrate the value of a reasoning-centered VLM backbone that unifies semantic perception, prediction, planning, and behavior abstraction. While the structured reasoning outputs of DriveRX can further benefit downstream planning or control modules, such extensions are beyond the scope of this work and are left for future research.

## B ETHICAL STATEMENT

This study involving vision - language models (VLMs) and large language models (LLMs) in the context of autonomous driving adheres to the following ethical principles:

- **Data Privacy and Legal Compliance**: All datasets used are sourced legally. Personal or sensitive information within the data is properly anonymized to protect individuals' privacy.

- **Research Transparency**: We are transparent about the data sources, research methods, and model limitations. The capabilities and limitations of our models are fully disclosed without exaggeration or concealment.

- **Scientific Integrity**: We maintain the highest standards of scientific integrity. Our research findings are based on rigorous experiments and objective analysis. Data fabrication or falsification is strictly prohibited.

- **Bias Mitigation**: We strive to minimize bias and discrimination in our models. We are committed to developing models that are fair and equitable.

- **Intellectual Property Respect**: We respect intellectual property rights and comply with relevant laws and regulations. Unauthorized use of proprietary data or models is avoided.

Our research focuses on exploring the potential of VLMs and LLMs in autonomous driving primarily from a theoretical and technical perspective. The emphasis is on improving the models' generalization capabilities and interpretability, which could provide valuable insights for future research and development in the field.

## C DATA COMPOSITION

We conducted 8 inference runs for Align-DS-V, Qwen2.5-VL-7B-Instruct, and Qwen2.5-VL-72B-Instruct on the DriveLM train dataset. Following the method in Eq. 1, we computed the difficulty of frames rather than individual questions. Figure 4 shows the resulting difficulty distributions for each model. Next, we filtered out DriveLM samples with mutually exclusive options using specific rules and cleaned and balanced the dataset with methods like Action balancing.

To meet different requirements, we used Aligs-DS-V model scores to get frame scores, selected samples with filtered frame scores of $[25, 45]$, and randomly sampled 1,000 frames as **AutoDriveRL-Train**. For the 3 models, we took samples with filtered frame scores of $[10, 31]$, removed overlaps with the training set, and obtained **DriveLM-Hard**, which represents a more challenging subset.

The task distribution of **AutoDriveRL-Train** and **DriveLM-Hard** can be observed in Figure 5.

## D MODELS

**LLaVA-1.5** (Liu et al., 2024a) is an open-source multimodal vision-language model developed by the LLaVA team, designed to understand and generate language in response to visual inputs. It integrates a CLIP-based vision encoder with a Vicuna (Zheng et al., 2023) language model through a multi-layer perceptron (MLP) connector, enabling it to perform tasks such as visual question answering, image captioning, and optical character recognition.

**LLaVA-NeXT** (Liu et al., 2024b) is an advanced multimodal model based on LLaVA-1.5, developed to enhance performance in various applications. It demonstrates improved capabilities in reasoning, OCR, and world knowledge. The model supports higher image resolutions, utilizes refined datasets, and is built on more powerful language model foundations.

**Qwen2.5-VL** (Bai et al., 2025) is a multimodal model family developed by Alibaba Cloud, capable of handling tasks such as visual question answering, image/video captioning, and document understanding. It demonstrates strong performance across diverse vision-language applications.

**InternVL3** (Zhu et al., 2025) is a multimodal large language model family developed by Shanghai Artificial Intelligence Laboratory. It features a native multimodal pre-training approach, integrating text, image, and video modalities within a unified framework.

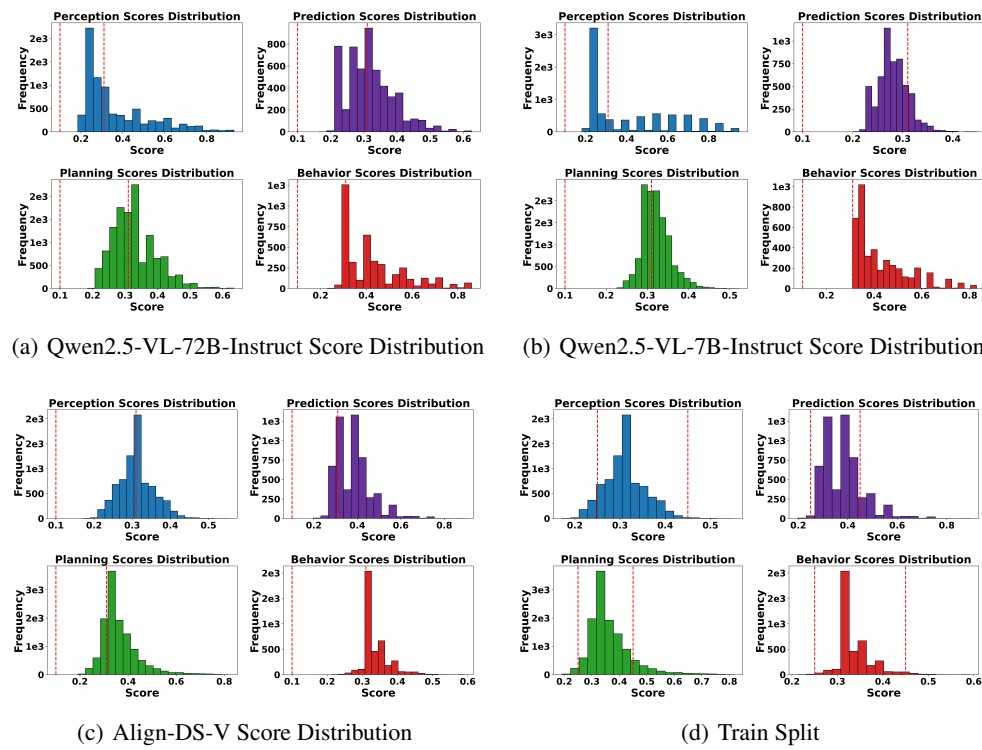

Figure 4: The model score distribution based on DriveLM-Hard

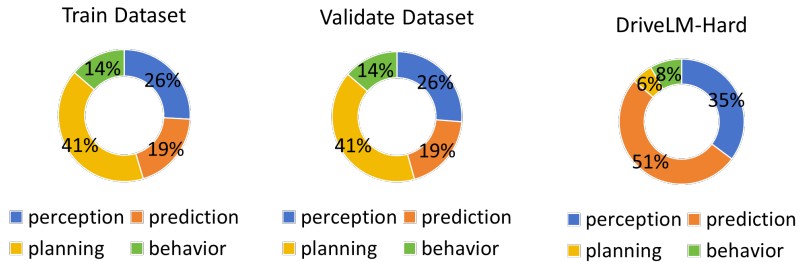

Figure 5: The proportion of each task in the dataset

**MM-Eureka** (Meng et al., 2025) is a multimodal reasoning model extends large-scale rule-based reinforcement learning (RL) to multimodal reasoning.

**R1-Onevision** (Yang et al., 2025b) is a multimodal reasoning model designed to bridge the gap between visual perception and deep reasoning.

**Align-DS-V** (PKU-Alignment, 2025) is an experimental vision-language model that extends the DeepSeek-R1-Distill-Llama-8B, focusing on enhancing reasoning capabilities through all-modality alignment. It demonstrates strong performance on visual question answering and mathematical reasoning tasks.

**DriveLM** (Sima et al., 2024) is a vision-language model designed for end-to-end driving systems, developed by OpenDriveLab. It integrates large-scale vision-language models (VLMs) into driving systems to enhance generalization and interaction with human users.

**Dolphin** (Ma et al., 2024) is a multimodal vision-language model tailored for autonomous driving, built upon OpenFlamingo, and enhanced via Grounded Chain-of-Thought and in-context instruction tuning to achieve human-like understanding, reasoning.

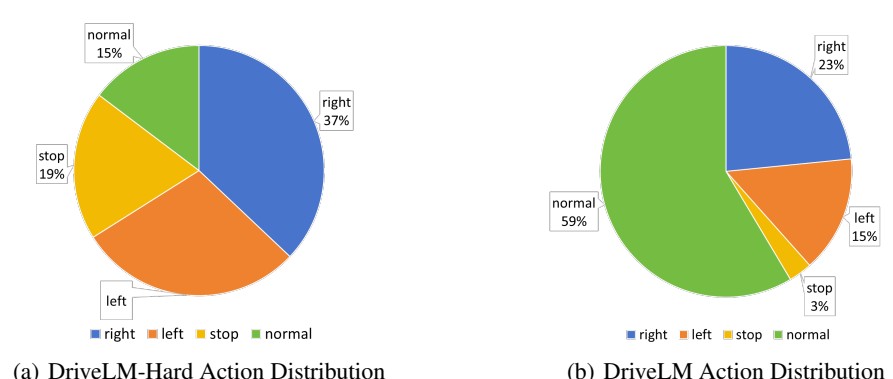

(a) DriveLM-Hard Action Distribution (b) DriveLM Action Distribution

Figure 6: The Action distribution on DriveLM-Hard and DriveLM

**GPT-4o** (OpenAI, 2024a) is a unified multimodal model by OpenAI that processes text, image, and audio inputs natively, enabling fast and accurate cross-modal reasoning with low latency.

## E   BENCHMARK

**DriveBench** (Xie et al., 2025) is a benchmark dataset designed to assess the reliability of Vision-Language Models (VLMs) for autonomous driving. It consists of 19,200 frames and 20,498 question-answer pairs, covering a wide range of settings and tasks. It's constructed by extracting a subset of frames from DriveLM-Train. Its innovation lies in being specifically created for evaluating the reliability of VLMs in autonomous driving scenarios, with diverse settings offering a comprehensive evaluation of VLMs' performance in practical applications.

**DriveLM** (Sima et al., 2024) is a method that uses a visual language model (VLM) to boost autonomous driving capabilities. It presents the Graph VQA task to replicate the multi-step reasoning of human drivers. Based on nuScenes and CARLA, the DriveLM dataset is created. The DriveLM-Agent, a VLM-based baseline approach, is proposed to perform Graph VQA and end-to-end driving jointly. Its innovation is introducing the Graph VQA task to integrate visual understanding, language interaction, and driving decisions, offering a new approach for autonomous driving development.

**Bench2Drive** (Jia et al., 2024) is a benchmark for evaluating end-to-end autonomous driving systems in closed-loop scenarios. It introduces 220 short routes covering 44 interactive driving scenarios, including varying weather and environmental conditions. Based on CARLA, the dataset provides 2 million annotated frames for fair and comprehensive evaluation. Bench2Drive allows for detailed testing of driving capabilities such as trajectory prediction and action execution, enabling the development of more robust autonomous driving models.

**DriveAction** (Hao et al., 2025) is the first action-driven benchmark specifically designed for Vision-Language-Action (VLA) models in autonomous driving. It comprises 16,185 QA pairs from 2,610 real-world driving scenarios. The benchmark uses driver-contributed real-world data and high-level discrete action labels collected directly from real-time driver operations. Its innovation is the action-rooted, tree-structured evaluation framework, which systematically links vision, language, and action tasks (V-L-A). This framework is used to rigorously evaluate state-of-the-art Vision-Language Models (VLMs) across different V-L-A modes, revealing that both vision and language guidance are necessary for accurate action prediction.

## F   IMPACT OF VISION ENCODERS ON PERCEPTION TASKS

We conduct a statistical analysis on the impact of different vision encoders on the **Perception** task. In addition to performance scores, we also compare each model's average response length and the number of image tokens generated by their visual encoders, as summarized in Table 5.

We observe two main findings. First, the capacity and design of the vision encoder significantly affect perception performance. Larger and more advanced encoders consistently achieve higher scores. For example, **InternVL3 78B** (with InternViT-6B-448px-V2.5) achieves a DriveBench perception score of 39.44, substantially outperforming its smaller variant **InternVL3 8B** (with InternViT-300M-448px-

Table 5: Impact of different vision encoders on the perception task. We report DriveBench perception and behavior scores (higher is better), average response length (in tokens) per question, number of image tokens per image, and inference time per question (in seconds). For a fair comparison, inference time is reported only among models of the same size. DriveLM requires 4 inference steps.

| Model | Size | 🖥 Vision Encoder | 🐢 Response | 🖼 Image Tokens | 🖼 Perception (↑) | 🖼 Behavior | ⏱ Infer times |
|---|---|---|---|---|---|---|---|
| MM-Eureka | 7B | Qwen-ViT-L/14 | 543.12 | 1824 | 34.44 | 49.40 | 2.52 |
| R1-Onevision | 7B | Qwen-ViT-L/14 | 447.55 | 1824 | 28.64 | 41.57 | 2.59 |
| Qwen2.5-VL | 7B | Qwen-ViT-L/14 | 357.85 | 1824 | 31.92 | 47.40 | 2.76 |
| Qwen2.5-VL | 72B | Qwen-ViT-L/14 | 343.27 | 1824 | 37.56 | 55.57 | – |
| InternVL3 | 8B | InternViT-300M-448px | 270.94 | 2304 | 33.21 | 51.47 | 1.58 |
| InternVL3 | 78B | InternViT-6B-448px | 275.69 | 2304 | 39.44 | 50.73 | – |
| DriveLM | 7B | CLIP ViT-L/14 | 623.04 | 576 | 16.85 | 42.78 | 3.18 |
| Align-DS-V | 8B | CLIP ViT-L/14 | 252.10 | 576 | 31.41 | 50.53 | 0.68 |
| **DriveRX** | 8B | CLIP ViT-L/14 | 363.44 | 576 | 32.88 | 62.02 | 1.06 |

V2.5), which only achieves 33.21. Furthermore, early ViT models like CLIP-ViT-L/14 underperform in perception tasks relative to newer large-scale ViT models such as InternViT-300M-448px-V2.5.

Second, the visual encoder plays a critical role in determining the upper bound of perception task performance during training. In our experiments, the improvement of **DriveRX** over its base model **Align-DS-V** is limited (+1.47), despite a substantial gain on the behavior task (+11.49). As shown in the training curves (Figure 5(b)), the perception reward for DriveRX saturates early (within 50 steps) and remains below 50, indicating a clear performance ceiling. We attribute this to the outdated architecture of CLIP-ViT-L/14, an early-generation ViT model whose limited perceptual capability falls behind modern ViT-based encoders.

However, high-performing models like Qwen2.5-VL (Team, 2024) and InternVL3 (Zhu et al., 2025) typically adopt dynamic resolution mechanisms, which allow them to process high-resolution inputs more effectively. While this improves visual understanding and perception scores, it also leads to significantly longer image token sequences. For instance, Qwen2.5-VL generates substantially more vision tokens per image compared to fixed-resolution VLMs like Align-DS-V. This increase in token length introduces notable inference latency, posing challenges for real-time decision-making in autonomous driving. In contrast to these models, **DriveRX** incorporates a fixed-length feature processing pipeline in its architectural design. While this design sacrifices some perceptual performance due to the output dimensionality constraints of the CLIP encoder, it achieves three critical advantages for practical deployment: deterministic inference efficiency, faster processing speed, and more predictable inference behavior.

# G ABLATION STUDY

## G.1 ABLATION OF SUBTASKS

To further verify the effectiveness of joint training, we conducted additional ablation studies on the DriveLM-Hard benchmark by withholding training data for each individual subtask. As shown in Table 6, removing any of the four subtasks (*Perception*, *Prediction*, *Planning*, or *Behavior*) consistently degraded the final behavior score, highlighting that each task provides indispensable information for downstream decision-making. We also evaluated a sequential pipeline where the output of each task is directly cascaded into the next without joint training, and found that its performance was similarly unsatisfactory. This suggests that the benefit of our framework arises not only from the decomposition itself but also from the integrated optimization across subtasks, as the interdependence between earlier and later tasks is crucial for accurate reasoning.

## G.2 ABLATION OF DATA FILTERING

Additionally, we include a controlled ablation to isolate the effect of our data-filtering strategy. Specifically, we randomly sample from the raw dataset the same number of training instances as in the filtered dataset, keeping all training configurations identical, and evaluate on DriveLM-Hard.

Table 6: Ablation study on the DriveLM-Hard benchmark. We report task-wise VQA scores for each setting. Removing any subtask or replacing joint training with sequential cascading leads to clear performance degradation.

| Model | Perception | Prediction | Planning | Behavior |
|---|---|---|---|---|
| w/o Perception | 25.13 | 57.81 | 37.12 | 27.15 |
| w/o Prediction | 27.41 | 42.12 | 41.25 | 30.95 |
| w/o Planning | 27.21 | 60.43 | 34.05 | 31.20 |
| Sequential Cascade | 29.45 | 49.45 | 26.53 | 29.45 |
| Align-DS-V | 29.45 | 52.90 | 29.15 | 32.48 |
| DriveRX (Ours) | **34.83** | **71.84** | **51.42** | **36.82** |

Table 7: **Ablation Study on the Effect of Data Filtering in RL Training.** We compare the performance of DriveRX trained with filtered and unfiltered data across multiple tasks on the DriveLM-Hard benchmark. The table highlights the impact of data filtering on model performance and stability.

| Model | Perception | Prediction | Planning | Behavior |
|---|---|---|---|---|
| Align-DS-V (base) | 29.45 | 52.90 | 29.15 | 32.48 |
| DriveRX w/ Data Filtering | 34.83 | 71.84 | 51.42 | 36.82 |
| DriveRX w/o Data Filtering | 30.29 | 51.82 | 48.23 | 33.07 |

As shown in Table 7, the model trained on the filtered corpus outperforms the unfiltered baseline across all four core tasks on DriveLM-Hard. This consistent performance advantage across tasks demonstrates that our data-filtering strategy provides cleaner, more task-aligned supervision signals, which in turn enhance training stability and final model performance in RL.

Furthermore, the score curves on the validation set during training highlight the impact of data filtering. As shown in the figure 7 below, the model trained on the filtered data (blue curve) exhibits (i) much faster improvement on the validation set from the very first steps, and (ii) consistently higher performance compared to the model trained on randomly sampled data (red curve). The test curves, evaluated using the same test set, clearly illustrate the faster convergence and superior performance of the filtered-data model, emphasizing the advantages of our data-filtering strategy.

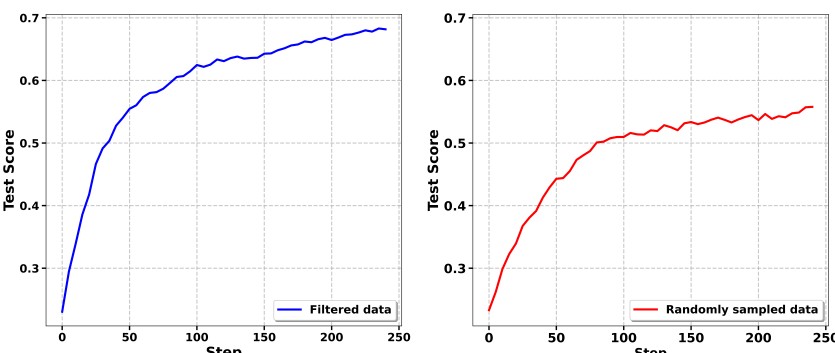

Figure 7: Score curves on the validation set during training.

## H  DATA DISTILLATION DETAILS

All training data for both **DriveRX-Agent** and **DriveRX-VLA** are distilled from the DriveRX reasoning corpus. We start from two original datasets processed by DriveLM (Sima et al., 2024): the nuScenes (Caesar et al., 2020) dataset for real-world scenes and the CARLA (Dosovitskiy et al., 2017) dataset for simulation. For each sample, we input the original prompt and ground-truth answer

into DriveRX to generate an intermediate reasoning trace encapsulated in the field, which provides structured decision-making information beyond the final output.

Based on this reasoning-enriched data, we construct two types of supervision signals. For DriveRX-Agent, the ground-truth vehicle trajectory is projected into a discrete vocabulary space by partitioning continuous coordinates into 512 bins, each represented by a unique token. These trajectory tokens serve as supervision targets, enabling the model to directly output discrete waypoint sequences conditioned on camera observations, ego-vehicle states, and context. Case study in figure 17.

For DriveRX-VLA, we cluster short motion segments using a K-means algorithm to obtain 2048 discrete action tokens, each representing a short-term executable maneuver such as steering, acceleration, or braking. These tokens are incorporated into the model's vocabulary, enabling the VLA model to directly predict control actions during inference.

During inference, both models receive multi-modal inputs, including camera images and ego-vehicle state information, and generate either trajectory tokens (DriveRX-Agent) or action tokens (DriveRX-VLA). These tokens are then decoded into continuous trajectories or executable control signals. This unified data distillation and tokenization pipeline ensures that both models learn from structured reasoning traces and ground-truth supervision, providing strong generalization and control capabilities.

# I ALIGNMENT BETWEEN HUMAN AND MODEL REWARDS

To reduce the cost of API-based evaluation, we use **Qwen2.5-72B-Instruct** as the default reward model. To assess the validity of using open-source LLMs for reward estimation, we conduct a correlation analysis between model scores and human judgments.

We design an online annotation platform for human evaluation and randomly select 300 samples from the evaluation set. Each sample is independently rated by three annotators. To ensure consistency, all annotators follow the same rubric used by our reward models, including task-specific criteria such as correctness, reasoning, and action relevance (see Appendix M). The average of the three scores is treated as the human gold score.

We then compute the Pearson correlation coefficient between human scores and model scores across several judge models, including the GPT-4 family (GPT-4o, GPT-4-turbo, GPT-3.5) and the Qwen2.5 family (7B and 72B).

The GPT-4o correlation (r = 0.83) demonstrates a strong alignment between model-based reward estimation and human judgment. Notably, this level of agreement matches the inter-annotator consistency reported for nuScenes-AP@0.5: when two human annotators draw bounding boxes, AP@0.5 considers them a match if their boxes overlap by at least 50% IoU (nuScenes (Caesar et al., 2020), App. A4). In other words, GPT's judgment is about as consistent with a human grader as two human annotators are with each other under that widely accepted 0.5-IoU criterion.

Table 8: Pearson correlation between human scores and model scores.

| Judge Model | GPT-4o | GPT-4t | GPT-4 | GPT-3.5 | Qwen2.5-72B | Qwen2.5-7B |
|---|---|---|---|---|---|---|
| **Pearson w/ Human** | 0.8309 | 0.7473 | 0.7473 | 0.6563 | 0.8138 | 0.4956 |

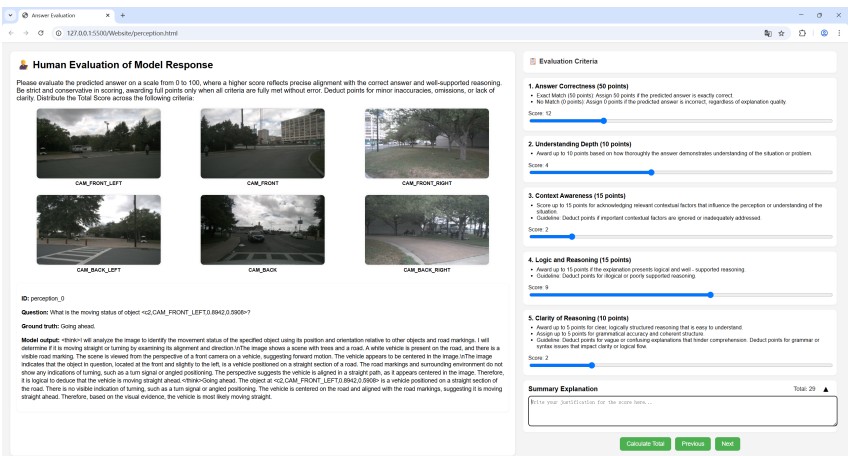

Figure 8: The scoring website used by human evaluators. Case 🧑‍🤝‍🧑 **Perception**

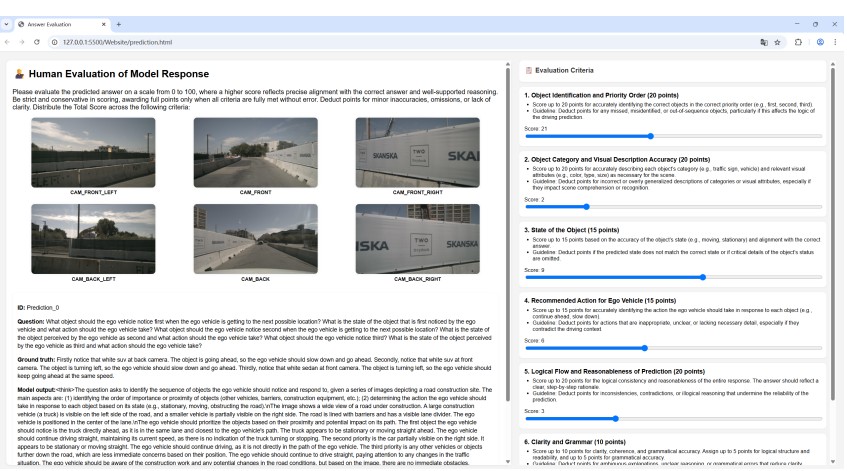

Figure 9: The scoring website used by human evaluators. Case 🚸 **Prediction**

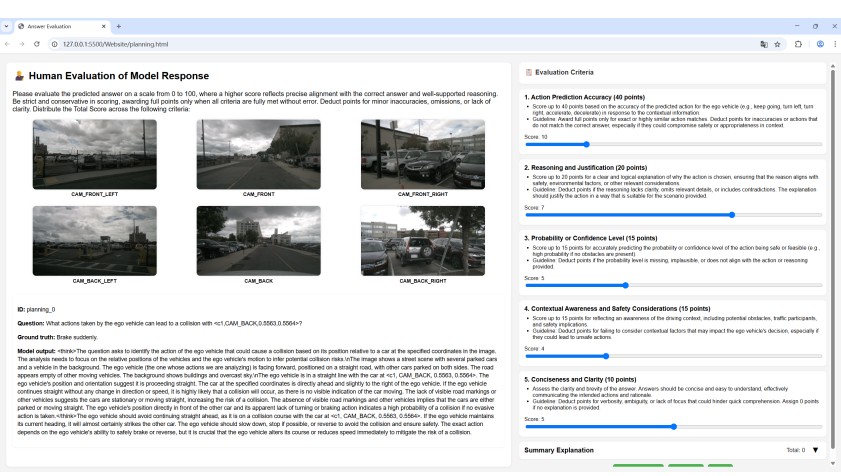

Figure 10: The scoring website used by human evaluators. Case 🏁 **Planning**

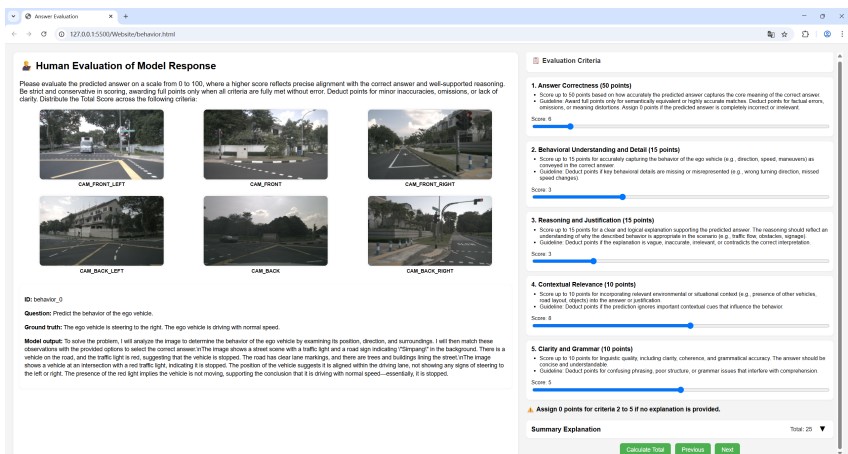

Figure 11: The scoring website used by human evaluators. Case 🧑‍🦰 **Behavior**

## J  MORE EVALUATION METRICS

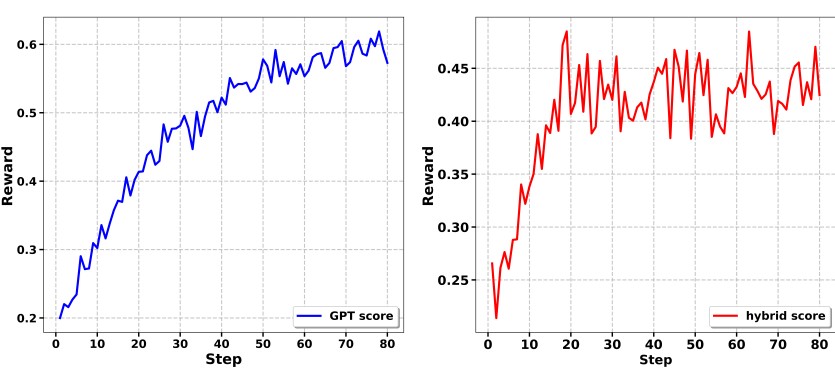

Figure 12: Training curves of GPT-based vs. hybrid reward scoring.

Evaluation of autonomous driving reasoning systems must be highly task-specific, as each sub-task emphasizes distinct aspects of perception and decision-making. In our framework, we design four dedicated evaluation prompts, corresponding to *Perception*, *Prediction*, *Planning*, and *Behavior*, ensuring that each score measures the attributes most relevant to its target task. For example, *Perception* requires not only identifying critical objects but also providing their spatial locations in structured coordinate form.

Conventional metrics often fail to capture the full complexity of such tasks. IoU-based metrics become unreliable when the predicted and reference object lists differ in number or ordering, and n-gram-based text similarity measures (e.g., BLEU, ROUGE) penalize valid paraphrases that nonetheless convey the correct semantics. These limitations motivate the use of GPT-based scoring, which can jointly consider semantic correctness, reasoning quality, and structural alignment in a unified evaluation.

To further investigate alternative evaluation strategies, we designed a hybrid reward scheme that integrates rule-based, semantic, and structural metrics, tailored to different question types:

- **Classification tasks:** For questions with discrete options or yes/no answers, we apply rule-based scoring, assigning 1 point for a correct match with the ground truth and 0 otherwise.
- **Open-ended tasks:** For free-form answers, we compute semantic similarity between predictions and references using GPT-based scoring, capturing meaning beyond surface form.
- **Structured text + coordinate tasks:** We separately evaluate textual descriptions with BLEU and ROUGE$_L$ metrics, while assessing coordinate precision independently.

- **Coordinate-only tasks:** We extract all floating-point numbers from the predicted and reference outputs, group them into coordinate pairs, and compute an F1 score by considering a prediction correct if the absolute difference between corresponding coordinates is less than 16 pixels.

Although the hybrid approach provides more fine-grained supervision, it introduces significant instability during reinforcement learning. Empirically, we observe that reward curves fluctuate heavily and are difficult to stabilize, in contrast to the smooth and monotonic improvements achieved with pure GPT-based scoring. Such instability may result from inconsistencies in metric scales, conflicts between optimization objectives, or the high sensitivity of structured matching signals to small output variations. As shown in Figure 12, even the best-performing hybrid runs exhibit large oscillations and lack a clear convergence trend.

These findings highlight a fundamental trade-off between metric granularity and training stability. Hybrid metrics offer more detailed signal decomposition but are highly susceptible to gaming behavior. GPT-based evaluation, by contrast, provides a holistic and robust measurement of task performance, balancing semantic understanding, structured reasoning, and action relevance. For this reason, we adopt GPT-based scores as our primary evaluation signal across all four task categories. Future work could explore dynamic weighting or staged evaluation to balance granularity and stability.

## K    COMPARISON WITH DIFFERENT TRAINING STRATEGIES

To validate the effectiveness of joint rl in AutoDriveRL, we conduct comparative experiments under the same data scale. Specifically, we compare (i) supervised fine-tuning (SFT), (ii) task-wise separate RL training (on perception, prediction, planning, and behavior), and (iii) our joint RL training strategy. All three variants are based on the same base model, Align-DS-V, and are evaluated on DriveBench. The results are shown in Table 9

Table 9: **Performance Evaluation of Diverse Training Strategies on Align-DS-V Across Multiple Driving Tasks in DriveBench** Each column shows the GPT score ($\uparrow$) for the clean version of each task. We  highlight  the best-performing open-source model for each task.

| Method | Size | 🧑 Perception | 🚥 Prediction | 🏗 Planning | 🎮 Behavior |
|---|---|---|---|---|---|
| DriveRX-SFT | 8B | 23.73 | 43.07 | 73.07 | 15.34 |
| DriveRX-Separate | 8B | 24.73 | 46.59 | 76.11 | 55.18 |
| DriveRX-JointRL | 8B | 32.88 | 52.20 | 78.63 | 62.02 |

In Table 9, we observe that the joint RL training scheme significantly improves performance across all four tasks under the same data scale. This result confirms that jointly optimizing multiple reasoning stages—rather than training them in isolation—enhances the model's overall effectiveness in autonomous driving tasks.This highlights the benefit of jointly optimizing semantically distinct reasoning stages, allowing the model to learn shared representations and task interactions that improve generalization. Furthermore, Table 9 shows that, on the behavior task, the performance of the model trained with SFT is significantly lower than that of the model trained with RL. This substantial gap arises because driving behavior requires **long-chain reasoning** (Perception→Prediction→Planning→Behavior), rather than simple pattern recognition. SFT, which mimics static, non-reasoning annotations, struggles to internalize the complex, multi-step logic necessary for high-level decision-making. In contrast, RL with Group Relative Policy Optimization (GRPO) enables the model to explicitly explore multiple reasoning paths, learning "how to think" rather than just "what to say." Furthermore, the complexity of autonomous driving, with its intricate logic and safety constraints, amplifies the SFT-RL gap. Unlike simpler VQA tasks, where this gap is minimal, the reasoning-intensive nature of behavior generation in driving tasks highlights the disproportionate benefits of RL's exploration and self-correction capabilities.

## L    CASE STUDY

Below is our Case Study, with Fig. 13, Fig. 14, Fig. 15, and Fig. 16 corresponding to the tasks of **Perception**, **Prediction**, **Planning**, and **Behavior** respectively.

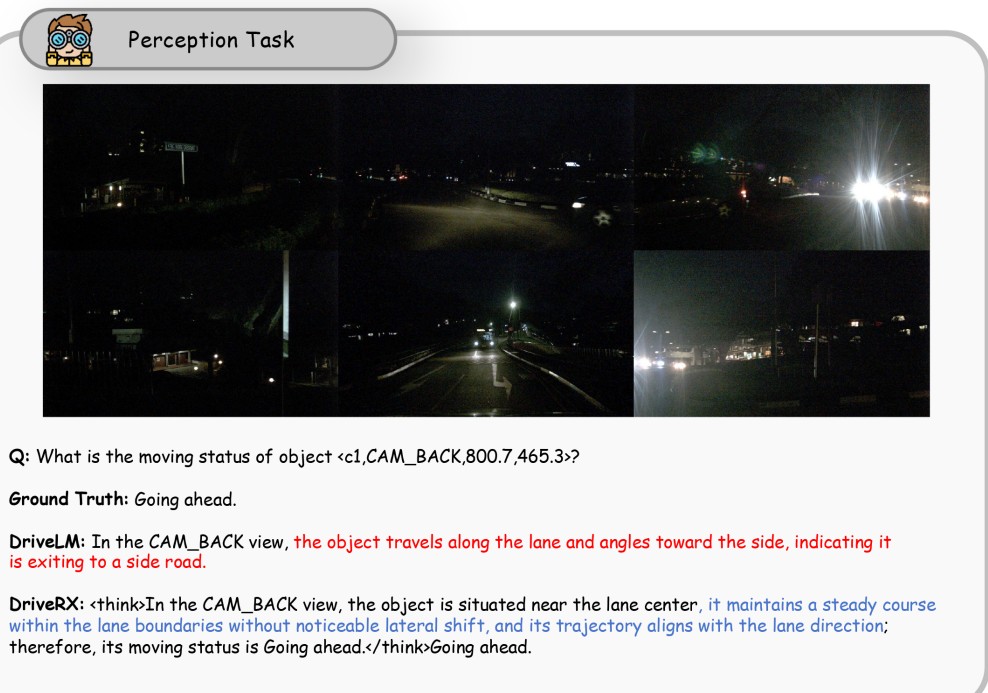

Figure 13: Perception task example comparing DriveLM and DriveRX. The question asks for the moving status of an object in the rear camera view (CAM_BACK). DriveLM hallucinates visual features, incorrectly stating that the object angles toward the side (highlighted in red). In contrast, DriveRX utilizes internal reasoning to correctly analyze the trajectory, noting that the object maintains a steady course without noticeable lateral shift (highlighted in blue), matching the Ground Truth "Going ahead".

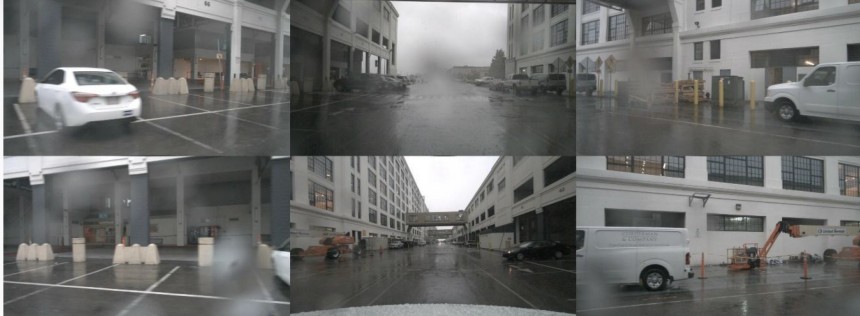

**Prediction Task**

**Q:** Is <c1,CAM_FRONT,394.2,496.7> a traffic sign or a road barrier?

**Ground Truth:** No.

**Qwen2.5-VL-72B-Instruct:** The object at <c1,CAM_FRONT,394.2,496.7> appears to be a road barrier. It is positioned in the middle of the road and has a structure consistent with a barrier used to block or guide traffic. Traffic signs are typically mounted on poles or posts and are not placed directly in the middle of the road. The object's placement and structure suggest it is a barrier rather than a sign.

**DriveRX:** <think>I will analyze the image for elements that resemble a traffic sign or road barrier at the specified coordinates. I will follow a structured approach to identify and describe these elements and determine if they match the description provided.
The image shows a wide view of an empty parking area in front of a large building. The area is wet with rain, and there are several parked vehicles and visible structures. There are no visible traffic signs or barriers in the foreground.
To determine if there is a traffic sign or road barrier at the specified coordinates <c1,CAM_FRONT,394.2,496.7>, I will examine the image closely. The coordinates suggest a specific location on the image. I will look for any visible objects or structures at this point that could resemble a traffic sign or road barrier. In this image, the area specified appears empty, and no such structures are visible.</think>
No. Based on the image, there is no visible traffic sign or road barrier at the specified coordinates. The image shows an empty parking area with no visible signage or barriers. Therefore, the statement that <c1,CAM_FRONT,394.2,496.7> is a traffic sign or a road barrier is incorrect.

Figure 14: Prediction task example comparing Qwen2.5-VL-72B-Instruct and DriveRX. The task is to determine if the object at the specified coordinates <c1,CAM_FRONT,394.2,496.7> is a traffic sign or a road barrier. Qwen2.5-VL-72B-Instruct incorrectly claims the object is a road barrier (highlighted in red), despite no such object being clearly visible at the coordinates, exhibiting object hallucination. In contrast, DriveRX correctly identifies that there is no visible traffic sign or road barrier at the specified coordinates (highlighted in blue) by performing a structured visual analysis of the indicated image region. This highlights DriveRX's improved ability to accurately perceive the scene and avoid hallucinations, which is critical for safe and reliable downstream driving decisions.

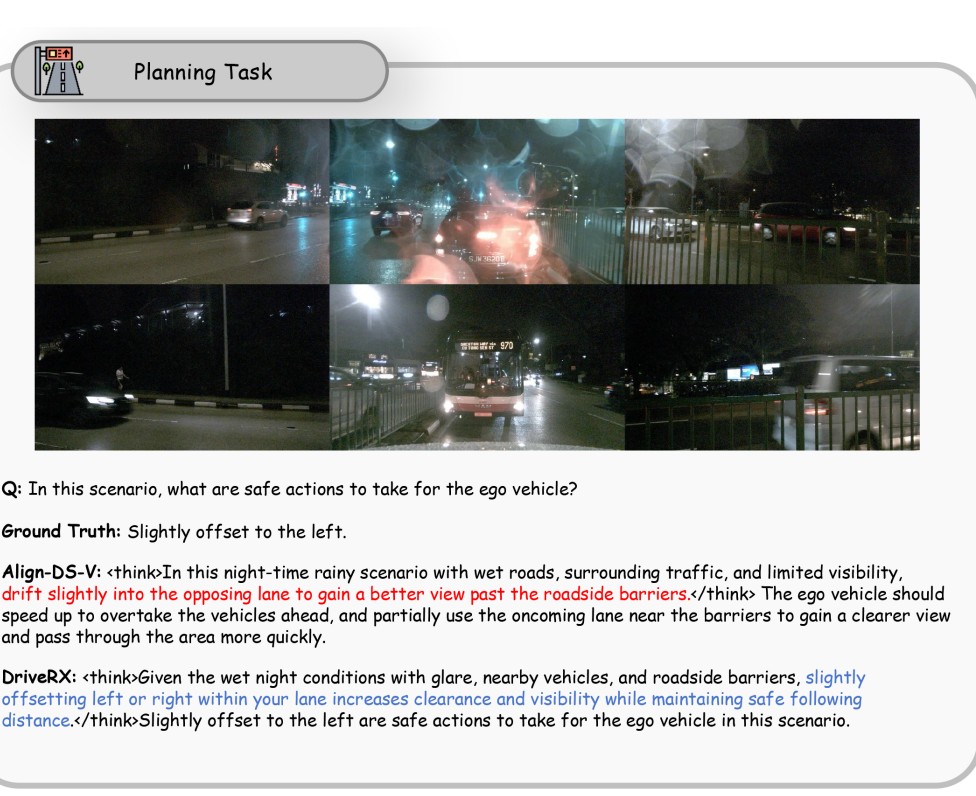

**Planning Task**

**Q:** In this scenario, what are safe actions to take for the ego vehicle?

**Ground Truth:** Slightly offset to the left.

**Align-DS-V:** \<think\>In this night-time rainy scenario with wet roads, surrounding traffic, and limited visibility, drift slightly into the opposing lane to gain a better view past the roadside barriers.\</think\> The ego vehicle should speed up to overtake the vehicles ahead, and partially use the oncoming lane near the barriers to gain a clearer view and pass through the area more quickly.

**DriveRX:** \<think\>Given the wet night conditions with glare, nearby vehicles, and roadside barriers, slightly offsetting left or right within your lane increases clearance and visibility while maintaining safe following distance.\</think\>Slightly offset to the left are safe actions to take for the ego vehicle in this scenario.

Figure 15: Planning task example comparing Align-DS-V and DriveRX. The goal is to determine safe actions for the ego vehicle. Align-DS-V suggests a potentially risky action by advising to drift into the opposing lane to gain a better view past roadside barriers (highlighted in red). In contrast, DriveRX provides a safer and more prudent recommendation, suggesting to slightly offset within the lane to increase clearance and visibility while maintaining a safe following distance (highlighted in blue). This comparison highlights DriveRX's ability to generate safer, lane-constrained planning decisions that align with the Ground Truth, avoiding the hazardous lane deviation proposed by Align-DS-V.

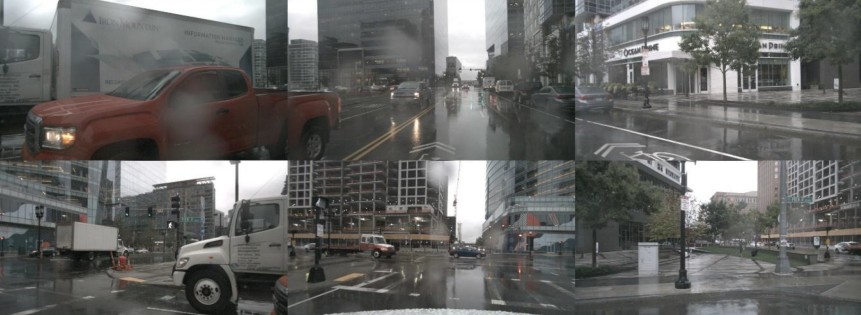

**Behavior Task**

**Q:** Predict the behavior of the ego vehicle.

**Ground Truth:** The ego vehicle is going straight. The ego vehicle is driving with normal speed.

**GPT-4o:**
Answer: The ego vehicle is steering to the left. The ego vehicle is driving slowly.
Explanation:
1.   Steering to the left: The images show the ego vehicle is positioned to make a left turn. The view from the front camera (CAM_FRONT) and the side cameras (CAM_FRONT_LEFT and CAM_FRONT_RIGHT) indicates the vehicle is aligning itself to the left, likely to follow the road or navigate around obstacles.
2.   Driving slowly: The wet road conditions, visible rain, and the presence of nearby vehicles and pedestrians suggest the ego vehicle is driving cautiously. Additionally, the lack of motion blur in the images supports the conclusion that the vehicle is moving at a slow speed.

**DriveRX:** <think>To solve the problem, I will analyze the image to determine the direction and speed of the ego vehicle by examining its position relative to the road markings and other vehicles. I will focus on the ego vehicle's alignment and movement indicators.
The image shows a street scene with a vehicle in the foreground. The ego vehicle, which is a white vehicle, is positioned on a wet road with visible road markings. The surrounding environment appears to be an urban setting with buildings and other vehicles visible in the background.
The ego vehicle is positioned in the center of the lane, aligned within the road markings. There are no visible indicators of sharp turning or rapid movement, such as angled positioning or blurred lines. The vehicle appears to be aligned straight, suggesting it is traveling straight. The surrounding conditions are wet, but there are no visual cues suggesting high speed. The ego vehicle's position and alignment within the lane support the conclusion that it is moving straight and at a steady speed.</think>
The ego vehicle is going straight. The ego vehicle is driving with normal speed.

Figure 16: Behavior prediction task example comparing GPT-4o and DriveRX. The goal is to predict the behavior of the ego vehicle in the current scene. GPT-4o incorrectly predicts that the ego vehicle is steering to the left, which could very likely result in a collision with the red vehicle on the left (highlighted in red). In contrast, DriveRX accurately concludes that the ego vehicle is going straight and driving with normal speed (highlighted in blue). DriveRX's analysis focuses on the vehicle's alignment within the lane markings and the absence of indicators for sharp turning or rapid movement, suggesting it is traveling straight. This task highlights the importance of accurate visual analysis and the ability to avoid incorrect assumptions in behavior prediction, with DriveRX demonstrating a more reliable and contextually appropriate approach.

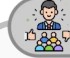 **Case Study of DriveRX-Agent**

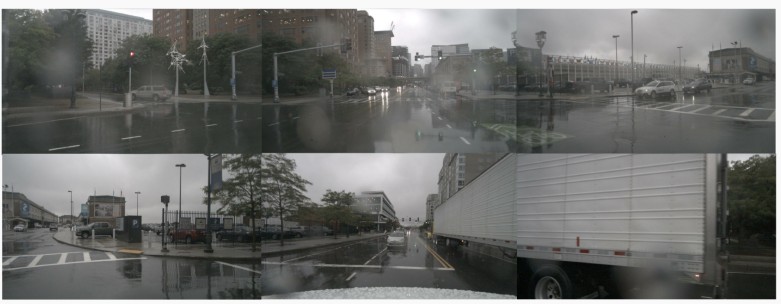

**Question:**
<image> You are an expert autonomous driving agent. Analyze the following scene description to plan a safe and efficient trajectory. The primary camera view corresponds to this description.\n\n### Ego Vehicle Status:\n- Current speed: <speed>...Task:Analyze the scene to determine the safest driving maneuver for the next 6 seconds.
Instructions:\n1. **Reasoning Process**: First, provide your detailed reasoning process inside the `<think>...</think>` tags. Your analysis should break down key information from the scene, such as traffic lights, other vehicles, and road conditions, without stating the final maneuver.\n2. **Maneuver Summary**: Following the `<think>` block, use one or two concise sentences to summarize the driving decision.\n3. **Trajectory Plan**: Finally, inside the `<trajectory>...</trajectory>` tags, provide a sequence of 12 waypoints (one every 0.5 seconds) based on your decision, represented by special location tokens.\n\nFormat:\n\n<think>\n[Your detailed reasoning]\n</think>\n[Your driving maneuver summary]\n<trajectory>[Your sequence of 12 waypoints]</trajectory>

**Model output:**
<think>To determine the ego vehicle's next behavior, I will analyze the image for visual cues indicating the vehicle's movement or intended action. ...</think>
The ego vehicle is going straight. The ego vehicle is driving slowly. <trajectory><loc_x_50><loc_y_127><loc_x_66><loc_y_103><loc_x_47><loc_y_129><loc_x_89><loc_y_131><loc_x_50><loc_y_85><loc_x_19><loc_y_143></trajectory>

Figure 17: Case study of DriveRX-Agent

# M  EXPERIMENT PROMPT

## M.1  TASK-SPECIFIC REWARD PROMPT

---

**👥 Reward prompt for Perception Tasks**

For Visual Question Answering questions:
Please evaluate the predicted answer on a scale from 0 to 100, where a higher score reflects precise alignment with the example correct answer and well-supported reasoning. Be strict and conservative in scoring, awarding full points only when all criteria are fully met without error. Deduct points for minor inaccuracies, omissions, or lack of clarity. Distribute the Total Score across the following criteria:

1. Information Completeness (30 points):
   - Score based on whether the predicted answer fully covers the key details in the example answer (including vehicle type, location, color, and object IDs). Deduct points for missing or incorrect information.

2. Detail Accuracy (30 points):
   - Assess whether the predicted answer accurately describes details like vehicle location and direction, matching the example answer. Points should be deducted for any deviations or ambiguous descriptions.

3. Clarity of Expression (20 points):
   - Evaluate whether the predicted answer is clearly and coherently expressed, allowing for accurate understanding of the information, similar to the example answer. Points should be deducted for confusing or ambiguous wording.

4. Format Compliance (20 points):
   - Check whether the predicted answer follows a similar format to the example answer, including punctuation and data presentation. Points should be deducted for non-compliant or inconsistent formatting.

Here is the correct answer: "{GT}"

Here is the predicted answer to be evaluated: "{PRED}"

Please fill in the following scoring sheet, and then provide a brief summary supporting the score:
1. Information Completeness (30 points):
2. Detail Accuracy (30 points):
3. Clarity of Expression (20 points):
4. Format Compliance (20 points):
Total Score:

Brief Summary:

---

Figure 18: The prompt guiding Qwen2.5-72B-Instruct to score the answers to visual question answering questions in perception task.

👥 **Reward prompt for Perception Tasks**

For Multiple-Choice questions:
Please evaluate the answer on a scale from 0 to 100, where a higher score reflects precise alignment with the correct response and well - supported reasoning. Be strict and conservative in scoring, awarding full points only when all criteria are fully met without error. Deduct points for minor inaccuracies, omissions, or lack of clarity. Distribute the Total Score across the following criteria:

1. Answer Correctness (50 points):
 - Exact Match (50 points): Assign 50 points if the predicted answer is exactly correct.
 - No Match (0 points): Assign 0 points if the predicted answer is incorrect, regardless of explanation quality.

2. Understanding Depth (10 points):
 - Award up to 10 points based on how thoroughly the answer demonstrates understanding of the situation or problem.

3. Context Awareness (15 points):
 - Score up to 15 points for acknowledging relevant contextual factors that influence the perception or understanding of the situation.
 - Guideline: Deduct points if important contextual factors are ignored or inadequately addressed.

4. Logic and Reasoning (15 points):
 - Award up to 15 points if the explanation presents logical and well - supported reasoning.
 - Guideline: Deduct points for illogical or poorly supported reasoning.

5. Clarity of Reasoning (10 points):
 - Award up to 5 points for clear, logically structured reasoning that is easy to understand.
 - Assign up to 5 points for grammatical accuracy and coherent structure.
 - Guideline: Deduct points for vague or confusing explanations that hinder comprehension. Deduct points for grammar or syntax issues that impact clarity or logical flow.

Here is the correct answer or solution: "{GT}"

Here is the predicted answer and explanation (if any): "{PRED}"

Please fill in the following scoring sheet, and then provide a brief summary supporting the score:
1. Answer Correctness (50 points):
2. Understanding Depth (10 points):
3. Context Awareness (15 points):
4. Logic and Reasoning (15 points):
5. Clarity of Reasoning (10 points):
Total Score:

Brief Summary:

Figure 19: The prompt for guiding Qwen2.5-72B-Instruct to evaluate the answers to multiple-choice questions in perception task.

**Reward prompt for Prediction Tasks**

For Visual Question Answering questions:
Please evaluate the predicted answer on a scale from 0 to 100, where a higher score reflects precise alignment with the correct answer and well-supported reasoning. Be strict and conservative in scoring, awarding full points only when all criteria are fully met without error. Deduct points for minor inaccuracies, omissions, or lack of clarity. Distribute the Total Score across the following criteria:

1. Object Identification and Priority Order (20 points):
- Score up to 20 points for accurately identifying the correct objects in the correct priority order (e.g., first, second, third) as indicated in the question.
- Guideline: Deduct points for any missed, misidentified, or out-of-sequence objects, particularly if this affects the logic of the driving prediction.

2. Object Category and Visual Description Accuracy (20 points):
- Score up to 20 points for accurately describing each object's category (e.g., traffic sign, vehicle) and relevant visual attributes (e.g., color, type, size) as necessary for the scene.
- Guideline: Deduct points for incorrect or overly generalized descriptions of categories or visual attributes, especially if they impact scene comprehension or recognition.

3. State of the Object (15 points):
- Score up to 15 points based on the accuracy of the object's state (e.g., moving, stationary) and alignment with the correct answer.
- Guideline: Deduct points if the predicted state does not match the correct state or if critical details of the object's status are omitted.

4. Recommended Action for Ego Vehicle (15 points):
- Score up to 15 points for accurately identifying the action the ego vehicle should take in response to each object (e.g., continue ahead, slow down).
- Guideline: Deduct points for actions that are inappropriate, unclear, or lacking necessary detail, especially if they contradict the driving context.

5. Logical Flow and Reasonableness of Prediction (20 points):
- Score up to 20 points for the logical consistency and reasonableness of the entire response. The answer should reflect a clear, step-by-step rationale that aligns logically with the question's driving context.
- Guideline: Deduct points for inconsistencies, contradictions, or illogical reasoning that undermine the reliability of the prediction.

6. Clarity and Grammar (10 points):
- Score up to 10 points for clarity, coherence, and grammatical accuracy. Assign up to 5 points for logical structure and readability, and up to 5 points for grammatical accuracy.
- Guideline: Deduct points for ambiguous explanations, unclear reasoning, or grammatical errors that reduce clarity.

Here is the question: "{QUESTION}"

Here is the correct answer: "{GT}"

Here is the predicted answer: "{PRED}"

Please fill in the following scoring sheet, and then provide a brief summary supporting the score:
1. Object Identification and Priority Order (20 points):
2. Object Category and Visual Description Accuracy (20 points):
3. State of the Object (15 points):
4. Recommended Action for Ego Vehicle (15 points):
5. Logical Flow and Reasonableness of Prediction (20 points):
6. Clarity and Grammar (10 points):
Total Score:

Brief Summary:

Figure 20: The prompt guiding Qwen2.5-72B-Instruct to score the answers to visual question answering questions in prediction task.

> ### 🧑‍🤝‍🧑 Reward prompt for Prediction Tasks
>
> For true/false questions:
> Please evaluate the predicted answer for the Yes/No question on a scale from 0 to 100, where a higher score reflects precise alignment with the correct answer and a well-supported explanation. Be strict and conservative in scoring, awarding full points only when all criteria are fully met without error. Deduct points for minor inaccuracies, omissions, or lack of clarity. Distribute the Total Score across the following criteria:
>
> 1. Answer Correctness (40 points):
> - Exact Match (40 points): Assign 40 points if the predicted Yes/No answer exactly matches the correct answer.
> - No Match (0 points): Assign 0 points if the predicted answer does not match the correct answer, regardless of explanation quality.
>
> 2. Object Category Identification (15 points):
> - Score up to 15 points for accurately identifying the object's category.
> - Guideline: Deduct points for any inaccuracies or missing elements in the category identification, particularly if they affect understanding or recognition of the object's role in the scene.
>
> 3. Object Visual Appearance (15 points):
> - Score up to 15 points for an accurate description of the object's visual appearance (e.g., colors, materials, size, or shape) as relevant to the question.
> - Guideline: Deduct points if any important visual details are missing, incorrect, or overly generalized, especially if they impact the explanation or perception of the object's function.
>
> 4. Object Position and Motion (15 points):
> - Score up to 15 points for correctly identifying the object's location, orientation, and motion (if applicable) relative to the ego vehicle.
> - Guideline: Deduct points for inaccuracies in spatial information, positioning, or motion. Include deductions if relevant motion or orientation details are omitted or incorrect.
>
> 5. Explanation Clarity and Justification (15 points):
> - Score up to 15 points for the clarity, logical structure, and justification of the explanation provided.
> - Guideline: Deduct points for vague, confusing, or insufficient explanations that fail to justify the Yes/No answer clearly and logically.
>
> Assign 0 points from criteria 2 to 5 if no explanation is provided.
>
> Here is the question: "{QUESTION}"
>
> Here is the correct answer: "{GT}"
>
> Here is the predicted answer and explanation (if any): "{PRED}"
>
> Please fill in the following scoring sheet, and then provide a brief summary supporting the score:
> 1. Answer Correctness (40 points):
> 2. Object Category Identification (15 points):
> 3. Object Visual Appearance (15 points):
> 4. Object Position and Motion (15 points):
> 5. Explanation Clarity and Justification (15 points):
> Total Score:
>
> Brief Summary:

Figure 21: The prompt for guiding Qwen2.5-72B-Instruct to evaluate the answers to judgment-type questions in prediction task.

**Reward prompt for Planning Tasks**

Please evaluate the predicted answer on a scale from 0 to 100, where a higher score reflects precise alignment with the correct answer and well-supported reasoning. Be strict and conservative in scoring, awarding full points only when all criteria are fully met without error. Deduct points for minor inaccuracies, omissions, or lack of clarity. Distribute the Total Score across the following criteria:

1. Action Prediction Accuracy (40 points):
- Score up to 40 points based on the accuracy of the predicted action for the ego vehicle (e.g., keep going, turn left, turn right, accelerate, decelerate) in response to the contextual information.
- Guideline: Award full points only for exact or highly similar action matches. Deduct points for inaccuracies or actions that do not match the correct answer, especially if they could compromise safety or appropriateness in context.

2. Reasoning and Justification (20 points):
- Score up to 20 points for a clear and logical explanation of why the action is chosen, ensuring that the reason aligns with safety, environmental factors, or other relevant considerations.
- Guideline: Deduct points if the reasoning lacks clarity, omits relevant details, or includes contradictions. The explanation should justify the action in a way that is suitable for the scenario provided.

3. Probability or Confidence Level (15 points):
- Score up to 15 points for accurately predicting the probability or confidence level of the action being safe or feasible (e.g., high probability if no obstacles are present).
- Guideline: Deduct points if the probability level is missing, implausible, or does not align with the action or reasoning provided.

4. Contextual Awareness and Safety Considerations (15 points):
- Score up to 15 points for reflecting an awareness of the driving context, including potential obstacles, traffic participants, and safety implications.
- Guideline: Deduct points for failing to consider contextual factors that may impact the ego vehicle's decision, especially if they could lead to unsafe actions.

5. Conciseness and Clarity (10 points):
- Assess the clarity and brevity of the answer. Answers should be concise and easy to understand, effectively communicating the intended actions and rationale.
- Guideline: Deduct points for verbosity, ambiguity, or lack of focus that could hinder quick comprehension. Assign 0 points if no explanation is provided.

Here is the question: "{QUESTION}"

Here is the ground truth object visual description: "{DESC}"

Here is the correct answer: "{GT}"

Here is the predicted answer: "{PRED}"

Please fill in the following scoring sheet, and then provide a brief summary supporting the score:
1. Action Prediction Accuracy (40 points):
2. Reasoning and Justification (20 points):
3. Probability or Confidence Level (15 points):
4. Contextual Awareness and Safety Considerations (15 points):
5. Conciseness and Clarity (10 points):
Total Score:

Brief Summary:

Figure 22: The prompt for guiding Qwen2.5-72B-Instruct to evaluate the answers to visual-related questions in planning task.

> ### 👨‍🏫 Reward prompt for Behavior Tasks
>
> Please evaluate the predicted answer on a scale from 0 to 100, where a higher score reflects precise alignment with the correct answer and well-supported reasoning. Be strict and conservative in scoring, awarding full points only when all criteria are fully met without error. Deduct points for minor inaccuracies, omissions, or lack of clarity. Distribute the Total Score across the following criteria:
>
> 1. Answer Correctness (50 points):
> - Exact Match (50 points): Assign 50 points if the predicted answer exactly matches the correct answer from the options.
> - No Match (0 points): Assign 0 points if the predicted answer does not match the correct answer, regardless of explanation quality.
>
> 2. Behavioral Understanding and Detail (15 points):
> - Score up to 15 points for accurately capturing the behavior details of the ego vehicle (e.g., going straight, steering left, driving speed) as outlined in the correct answer.
> - Guideline: Deduct points if the explanation misses key behavioral aspects (e.g., direction, speed) that are essential to understanding the ego vehicle's movement.
>
> 3. Reasoning and Justification (15 points):
> - Score up to 15 points for a clear and logical explanation justifying the chosen answer. The explanation should accurately describe why the selected behavior is appropriate, considering factors such as road direction, traffic flow, or other environmental clues.
> - Guideline: Deduct points if the reasoning lacks clarity, includes irrelevant details, or contradicts the behavior of the ego vehicle as described in the correct answer.
>
> 4. Contextual Relevance (10 points):
> - Score up to 10 points for the relevance of the explanation to the driving context, ensuring that it considers any environmental or situational factors that may influence the ego vehicle's behavior.
> - Guideline: Deduct points if the explanation fails to consider context, such as road conditions or nearby objects, that might impact the vehicle's behavior.
>
> 5. Clarity and Grammar (10 points):
> - Score up to 10 points for clarity, coherence, and grammatical accuracy of the explanation. The response should be concise and easy to understand.
> - Guideline: Deduct points for confusing language, vague statements, or grammatical errors that hinder comprehension.
>
> Assign 0 points from criteria 2 to 5 if no explanation is provided.
> Here is the question: "{QUESTION}"
>
> Here is the correct answer: "{GT}"
>
> Here is the predicted answer: "{PRED}"
>
> Please fill in the following scoring sheet, and then provide a brief summary supporting the score:
> 1. Answer Correctness (50 points):
> 2. Behavioral Understanding and Detail (15 points):
> 3. Reasoning and Justification (15 points):
> 4. Contextual Relevance (10 points):
> 5. Clarity and Grammar (10 points):
> Total Score:
>
> Brief Summary:

Figure 23: The prompt for guiding Qwen2.5-72B-Instruct to score the answers to multiple-choice questions in behavior task.

## M.2 INFERENCE PROMPT

The following prompt is used during our evaluation on the DriveLM-Hard benchmark.

**Inference prompt for DriveLM-Hard**

You are a smart autonomous driving assistant responsible for analyzing and responding to driving scenarios. You are provided with six camera images in the sequence [CAM_FRONT, CAM_FRONT_LEFT, CAM_FRONT_RIGHT, CAM_BACK, CAM_BACK_LEFT, CAM_BACK_RIGHT].

Instructions:
1. Answer Requirements:
- For multiple-choice questions, provide the selected answer choice along with an explanation.
- For "is" or "is not" questions, respond with a "Yes" or "No", along with an explanation.
- For open-ended perception and prediction questions, related objects to which the camera.

2. Key Information for Driving Context:
- When answering, focus on object attributes (e.g., categories, statuses, visual descriptions) and motions (e.g., speed, action, acceleration) relevant to driving safety and decision-making.

Use the images and coordinate information to respond accurately to questions related to perception, prediction, planning, or behavior, based on the question requirements.

{image_placeholders}

{Question}

Figure 24: The prompt guiding VLMs to answer DriveLM-Hard questions.

# N CORRUPTION DETAILS

We provide the detailed GPT scores, including corruption details, for different tasks in Tab. 10, Tab. 11, Tab. 12, Tab. 13. and Tab. 14

Table 10: Detailed GPT score results of **MCQs** for the 🧑 **Perception** task. "Clean" represents clean image inputs. The "Corrupt" settings range from weather conditions, external disturbances, sensor failures, motion blur, and transmission errors. The benchmarked VLMs include commercial, open-sourced, and driving specialist models, respectively.

| Method | Size | Clean | Brightness | Dark | Snow | Fog | Rain | Lens Obstacle | Water Splash | Camera Crash | Frame Lost | Saturate | Motion Blur | Zoom Blur | Bit Error | Color Quant | H.265 Compression |
|---|---|---|---|---|---|---|---|---|---|---|---|---|---|---|---|---|---|
| LLaVA-1.5 | 7B | 32.40 | 32.48 | 32.95 | 31.95 | 32.43 | 32.30 | 32.88 | 32.18 | 32.93 | 31.63 | 32.50 | 32.43 | 32.93 | 32.48 | 32.18 | 32.63 |
| LLaVA-NeXT | 7B | 32.98 | 33.85 | 20.43 | 11.62 | 16.58 | 27.33 | 18.24 | 32.30 | 26.80 | 20.83 | 23.50 | 34.00 | 17.75 | 24.50 | 18.03 | 26.24 |
| Qwen2.5-VL | 7B | 36.75 | 37.26 | 24.91 | 30.21 | 24.36 | 32.41 | 27.53 | 26.71 | 26.50 | 19.03 | 33.42 | 34.50 | 18.15 | 22.41 | 21.47 | 38.41 |
| InternVL3 | 8B | 36.22 | 39.97 | 35.94 | 42.59 | 37.24 | 41.41 | 41.24 | 39.39 | 33.74 | 26.38 | 35.06 | 39.82 | 41.33 | 31.00 | 40.65 | 42.12 |
| Qwen2.5VL | 72B | 47.68 | 39.71 | 26.35 | 38.59 | 42.56 | 45.03 | 38.41 | 39.38 | 28.21 | 19.38 | 44.18 | 39.85 | 36.76 | 32.35 | 29.55 | 49.47 |
| InternVL3 | 78B | 47.33 | 44.82 | 38.35 | 45.29 | 43.50 | 40.09 | 46.09 | 41.00 | 32.09 | 24.53 | 40.68 | 47.30 | 39.44 | 36.61 | 41.30 | 45.97 |
| MM-Eureka | 7B | 43.33 | 29.21 | 17.59 | 31.97 | 37.97 | 41.44 | 36.44 | 32.21 | 30.27 | 22.71 | 41.73 | 34.68 | 19.56 | 26.39 | 27.21 | 41.21 |
| R1-Onevision | 7B | 36.89 | 30.24 | 23.71 | 31.56 | 37.36 | 34.32 | 36.12 | 25.06 | 30.18 | 34.74 | 26.82 | 24.76 | 34.03 | 23.71 | 29.56 | 29.59 |
| DriveLM | 7B | 22.38 | 20.78 | 25.30 | 18.98 | 24.43 | 25.95 | 22.03 | 21.03 | 21.95 | 16.28 | 19.38 | 22.98 | 20.93 | 19.90 | 16.25 | 26.48 |
| Dolphin | 7B | 6.50 | 10.18 | 11.08 | 10.70 | 9.53 | 10.58 | 9.93 | 9.80 | 10.08 | 9.95 | 11.20 | 9.85 | 10.10 | 8.80 | 10.00 | 11.10 |
| Align-DS-V | 8B | 39.8 | 39.65 | 31.00 | 41.50 | 33.24 | 42.50 | 35.15 | 36.85 | 27.41 | 38.32 | 25.21 | 28.68 | 30.44 | 35.91 | | |
| DriveRX | 8B | 42.79 | 34.71 | 36.09 | 39.29 | 43.74 | 39.79 | 27.56 | 38.85 | 33.68 | 26.74 | 32.18 | 33.29 | 30.53 | 30.65 | 25.59 | 38.50 |

Table 11: Detailed GPT score results of **open-ended questions** for the 🧑 **Perception** task. "Clean" represents clean image inputs. The "Corrupt" settings range from weather conditions, external disturbances, sensor failures, motion blur, and transmission errors. The benchmarked VLMs include commercial, open-sourced, and driving specialist models, respectively.

| Method | Size | Clean | Brightness | Dark | Snow | Fog | Rain | Lens Obstacle | Water Splash | Camera Crash | Frame Lost | Saturate | Motion Blur | Zoom Blur | Bit Error | Color Quant | H.265 Compression |
|---|---|---|---|---|---|---|---|---|---|---|---|---|---|---|---|---|---|
| LLaVA-1.5 | 7B | 14.03 | 13.53 | 13.31 | 13.31 | 13.61 | 13.75 | 13.91 | 13.48 | 13.95 | 12.90 | 12.98 | 13.35 | 13.05 | 13.36 | 12.83 | 14.28 |
| LLaVA-NeXT | 7B | 15.33 | 15.49 | 14.95 | 16.61 | 15.62 | 16.05 | 15.66 | 15.66 | 15.19 | 16.04 | 15.16 | 16.06 | 17.92 | 15.27 | 15.10 | 15.86 |
| Qwen2.5-VL | 7B | 27.09 | 27.76 | 23.14 | 27.05 | 28.05 | 25.90 | 28.24 | 28.29 | 28.48 | 25.81 | 28.62 | 25.90 | 25.86 | 25.38 | 23.62 | 25.38 |
| InternVL3 | 8B | 30.21 | 30.38 | 31.29 | 32.33 | 28.52 | 29.57 | 28.95 | 31.33 | 29.95 | 25.29 | 30.76 | 29.76 | 30.67 | 28.71 | 29.90 | 30.33 |
| Qwen2.5VL | 72B | 27.43 | 29.29 | 26.76 | 23.86 | 25.05 | 29.38 | 28.14 | 28.43 | 29.38 | 26.14 | 27.86 | 28.71 | 28.10 | 27.52 | 29.48 | 29.71 |
| InternVL3 | 78B | 31.53 | 32.33 | 31.95 | 32.05 | 29.62 | 30.38 | 33.35 | 28.76 | 32.86 | 28.76 | 29.14 | 28.90 | 30.71 | 30.76 | 30.90 | 31.10 |
| MM-Eureka | 7B | 24.46 | 26.48 | 25.38 | 26.29 | 26.33 | 26.62 | 27.71 | 27.14 | 26.10 | 26.00 | 26.19 | 27.14 | 26.52 | 25.67 | 26.81 | 26.62 |
| R1-Onevision | 7B | 20.39 | 21.39 | 19.20 | 19.52 | 19.70 | 20.60 | 21.50 | 21.89 | 22.83 | 20.95 | 20.84 | 21.68 | 17.93 | 18.67 | 22.00 | 21.67 |
| DriveLM | 7B | 11.32 | 11.30 | 10.03 | 11.21 | 9.71 | 10.22 | 10.71 | 11.01 | 11.14 | 10.13 | 9.38 | 10.97 | 9.03 | 11.39 | 10.50 | 10.73 |
| Dolphin | 7B | 12.68 | 12.07 | 10.34 | 12.37 | 11.46 | 11.73 | 12.33 | 11.04 | 12.34 | 11.39 | 11.47 | 11.34 | 10.17 | 11.40 | 11.35 | 11.65 |
| Align-DS-V | 8B | 23.03 | 39.65 | 31.00 | 41.50 | 33.24 | 42.50 | 35.15 | 36.85 | 27.41 | 38.32 | 25.21 | 28.68 | 30.44 | 35.91 | 36.32 | 25.71 |
| DriveRX | 8B | 22.97 | 24.19 | 26.33 | 26.33 | 27.52 | 25.86 | 25.38 | 25.14 | 26.10 | 26.81 | 25.38 | 27.05 | 24.67 | 23.71 | 25.76 | 25.62 |

Table 12: Detailed GPT score results of the **open-ended questions** for 🚥 **Prediction**.

| Method | Size | Clean | Brightness | Dark | Snow | Fog | Rain | Lens Obstacle | Water Splash | Camera Crash | Frame Lost | Saturate | Motion Blur | Zoom Blur | Bit Error | Color Quant | H.265 Compression |
|---|---|---|---|---|---|---|---|---|---|---|---|---|---|---|---|---|---|
| LLaVA-1.5 | 7B | 22.02 | 24.79 | 20.95 | 15.97 | 15.30 | 18.98 | 16.28 | 24.16 | 6.11 | 13.90 | 20.30 | 25.20 | 10.56 | 11.10 | 15.61 | 23.92 |
| LLaVA-NeXT | 7B | 35.07 | 37.15 | 35.31 | 37.59 | 37.62 | 35.44 | 37.00 | 35.87 | 36.25 | 30.10 | 40.56 | 34.66 | 39.36 | 31.74 | 34.07 | 35.66 |
| Qwen2.5-VL | 7B | 42.26 | 50.75 | 36.58 | 47.25 | 46.00 | 47.58 | 54.50 | 53.67 | 45.00 | 46.17 | 44.83 | 48.08 | 43.00 | 48.00 | 43.25 | 52.25 |
| InternVL3 | 8B | 41.21 | 50.17 | 43.08 | 51.50 | 54.50 | 45.75 | 47.67 | 48.33 | 39.17 | 45.17 | 46.92 | 46.33 | 48.33 | 46.58 | 50.42 | 54.17 |
| Qwen2.5VL | 72B | 54.89 | 56.00 | 46.58 | 38.83 | 44.00 | 51.50 | 57.33 | 54.42 | 47.17 | 52.83 | 50.58 | 50.50 | 55.75 | 50.42 | 51.17 | 49.92 |
| InternVL3 | 78B | 50.93 | 55.50 | 44.92 | 51.50 | 59.00 | 53.92 | 51.67 | 50.33 | 46.92 | 42.83 | 49.08 | 47.75 | 51.42 | 45.08 | 47.75 | 48.00 |
| MM-Eureka | 7B | 40.31 | 44.42 | 37.58 | 36.00 | 36.75 | 32.42 | 42.91 | 45.42 | 45.17 | 53.75 | 45.33 | 43.92 | 39.75 | 39.17 | 38.75 | 44.50 |
| R1-Onevision | 7B | 51.22 | 44.40 | 33.25 | 47.92 | 58.17 | 47.75 | 46.33 | 47.42 | 48.80 | 46.25 | 42.33 | 46.27 | 47.45 | 52.08 | 45.10 | 50.18 |
| DriveLM | 7B | 44.33 | 46.82 | 43.90 | 42.33 | 35.84 | 44.13 | 44.00 | 42.59 | 46.25 | 33.56 | 29.69 | 42.15 | 19.00 | 38.20 | 44.33 | 42.87 |
| Dolphin | 7B | 32.66 | 29.85 | 32.31 | 24.64 | 29.92 | 31.38 | 33.41 | 31.79 | 29.05 | 30.93 | 30.49 | 31.59 | 26.38 | 30.13 | 25.64 | 30.62 |
| Align-DS-V | 8B | 41.51 | 41.50 | 35.83 | 38.75 | 32.67 | 39.42 | 42.50 | 38.33 | 41.92 | 32.83 | 42.92 | 44.42 | 35.25 | 40.67 | 35.42 | 35.00 |
| DriveRX | 8B | 52.20 | 51.42 | 41.75 | 48.83 | 48.92 | 47.17 | 49.25 | 51.92 | 47.92 | 38.67 | 55.42 | 48.17 | 53.25 | 48.50 | 39.50 | 53.75 |

Table 13: Detailed GPT score results of the **open-ended questions** for 🛣️ **Planning**.

| Method | Size | Clean | Brightness | Dark | Snow | Fog | Rain | Lens Obstacle | Water Splash | Camera Crash | Frame Lost | Saturate | Motion Blur | Zoom Blur | Bit Error | Color Quant | H.265 Compression |
|---|---|---|---|---|---|---|---|---|---|---|---|---|---|---|---|---|---|
| LLaVA-1.5 | 7B | 29.15 | 31.52 | 31.49 | 32.58 | 32.42 | 31.00 | 29.81 | 31.72 | 33.70 | 35.95 | 29.93 | 31.20 | 30.65 | 30.05 | 30.00 | 30.61 |
| LLaVA-NeXT | 7B | 45.27 | 45.64 | 44.54 | 43.55 | 44.17 | 45.08 | 44.62 | 44.21 | 45.69 | 44.51 | 41.17 | 45.30 | 44.57 | 43.67 | 43.57 | 45.13 |
| Qwen2.5-VL | 7B | 55.70 | 62.34 | 61.15 | 61.76 | 56.91 | 54.27 | 57.51 | 57.48 | 56.37 | 48.50 | 65.56 | 60.16 | 56.63 | 58.27 | 54.06 | 58.60 |
| InternVL3 | 8B | 70.97 | 54.07 | 51.07 | 50.00 | 53.82 | 55.96 | 49.00 | 49.57 | 54.77 | 52.09 | 50.56 | 49.07 | 48.49 | 51.88 | 49.99 | 52.94 |
| Qwen2.5VL | 72B | 77.18 | 80.95 | 69.67 | 75.22 | 76.24 | 77.79 | 78.12 | 78.40 | 69.98 | 53.11 | 74.74 | 80.84 | 75.13 | 73.33 | 68.98 | 77.67 |
| InternVL3 | 78B | 82.24 | 79.30 | 73.96 | 76.63 | 79.51 | 80.70 | 82.28 | 81.33 | 77.95 | 69.71 | 77.17 | 82.17 | 79.67 | 78.41 | 77.48 | 79.90 |
| MM-Eureka | 7B | 65.38 | 63.35 | 58.75 | 60.85 | 59.21 | 66.11 | 57.41 | 62.58 | 59.90 | 49.80 | 60.99 | 62.35 | 58.98 | 57.99 | 52.95 | 71.53 |
| R1-Onevision | 7B | 55.12 | 53.03 | 55.92 | 58.28 | 58.74 | 57.04 | 57.44 | 56.36 | 57.13 | 60.25 | 53.11 | 49.87 | 55.22 | 51.55 | 50.57 | 58.60 |
| DriveLM | 7B | 68.71 | 67.25 | 67.52 | 65.72 | 63.08 | 69.60 | 69.04 | 67.97 | 67.85 | 66.47 | 66.25 | 67.93 | 70.17 | 68.46 | 68.30 | 68.59 |
| Dolphin | 7B | 52.91 | 51.85 | 55.39 | 53.09 | 54.78 | 53.92 | 51.79 | 53.57 | 55.73 | 57.81 | 55.78 | 51.42 | 50.95 | 53.35 | 54.89 | 52.17 |
| Align-DS-V | 8B | 51.16 | 61.88 | 56.82 | 53.65 | 58.77 | 59.20 | 58.10 | 60.98 | 58.71 | 57.55 | 51.16 | 59.95 | 55.45 | 60.63 | 53.04 | 60.82 |
| DriveRX | 8B | 78.63 | 77.89 | 74.99 | 74.54 | 74.33 | 75.29 | 75.44 | 74.71 | 75.96 | 78.18 | 74.87 | 74.18 | 76.48 | 75.59 | 73.67 | 76.80 |

Table 14: Detailed GPT score results of the **MCQs** for 🚘 **Behavior**.

| Method | Size | Clean | Brightness | Dark | Snow | Fog | Rain | Lens Obstacle | Water Splash | Camera Crash | Frame Lost | Saturate | Motion Blur | Zoom Blur | Bit Error | Color Quant | H.265 Compression |
|---|---|---|---|---|---|---|---|---|---|---|---|---|---|---|---|---|---|
| LLaVA-1.5 | 7B | 13.60 | 12.79 | 12.83 | 15.57 | 12.63 | 14.06 | 13.99 | 12.79 | 14.68 | 13.65 | 13.12 | 13.55 | 13.83 | 13.98 | 13.44 | 13.48 |
| LLaVA-NeXT | 7B | 48.16 | 48.84 | 38.82 | 15.90 | 39.13 | 47.07 | 20.72 | 47.02 | 48.20 | 36.67 | 39.69 | 47.36 | 39.60 | 46.99 | 28.13 | 47.55 |
| Qwen2.5-VL | 7B | 47.40 | 50.59 | 30.63 | 46.11 | 45.26 | 46.30 | 36.26 | 37.85 | 54.56 | 49.41 | 44.81 | 39.56 | 36.81 | 33.59 | 48.89 | 41.67 |
| InternVL3 | 8B | 51.47 | 40.22 | 62.26 | 54.07 | 57.48 | 48.59 | 54.96 | 53.93 | 48.78 | 49.70 | 43.04 | 51.89 | 59.70 | 41.70 | 51.30 | 60.15 |
| Qwen2.5VL | 72B | 55.57 | 45.04 | 58.67 | 59.48 | 57.56 | 51.56 | 49.67 | 57.63 | 57.93 | 49.37 | 53.52 | 49.44 | 35.19 | 41.59 | 56.37 | 51.70 |
| InternVL3 | 78B | 50.73 | 47.70 | 56.04 | 42.41 | 55.78 | 53.07 | 50.67 | 51.41 | 59.30 | 51.19 | 68.19 | 42.85 | 37.04 | 54.07 | 58.59 | 52.15 |
| MM-Eureka | 7B | 49.4 | 47.58 | 54.05 | 36.73 | 39.04 | 59.79 | 46.11 | 51.45 | 50.86 | 61.00 | 42.00 | 56.93 | 30.61 | 46.48 | 34.47 | 56.80 |
| R1-Onevision | 7B | 41.57 | 39.35 | 57.17 | 38.31 | 31.74 | 48.33 | 45.15 | 51.23 | 44.00 | 37.07 | 37.92 | 30.81 | 32.04 | 35.85 | 32.26 | 36.93 |
| DriveLM | 7B | 42.78 | 47.18 | 36.30 | 40.70 | 39.18 | 40.93 | 43.30 | 40.98 | 39.95 | 38.23 | 40.08 | 45.68 | 38.88 | 41.10 | 33.50 | 39.65 |
| Dolphin | 7B | 8.81 | 7.17 | 9.54 | 9.02 | 6.48 | 8.05 | 7.95 | 7.10 | 9.29 | 8.94 | 8.02 | 8.02 | 9.42 | 8.37 | 10.07 | 6.32 |
| Align-DS-V | 8B | 50.53 | 50.26 | 36.07 | 38.04 | 38.67 | 44.78 | 28.52 | 54.41 | 48.19 | 34.30 | 30.96 | 36.22 | 35.56 | 52.33 | 34.52 | 50.63 |
| DriveRX | 8B | 62.02 | 55.67 | 61.04 | 54.07 | 57.15 | 60.35 | 56.22 | 54.70 | 58.63 | 51.19 | 46.85 | 57.30 | 41.15 | 45.88 | 69.04 | 63.48 |

## O  IMPLEMENTATION DETAILS

### O.1  HARDWARE CONFIGURATION

For all train and inference experiments, we utilized two computational cluster equipped with 8 NVIDIA A100-80GB GPUs. Our model DriveRX was trained across 16 NVIDIA A100-80GB GPUs for 5 epochs, a process that took 87 hours.

### O.2  HYPERPARAMETER SETTINGS

During GRPO (Shao et al., 2024) training, we adopt a prompt batch size of 128 is utilized, with 8 rollouts generated per prompt. The maximum rollout length is capped at 4,096 tokens. A default sampling temperature of 1.0 is set, and a KL loss coefficient of 1e-2 is employed. The training process consists of 5 epochs.

For Supervised Fine-tuning, we set the learining rate to 1e-6 trained for 3 epochs.

To ensure the reliability of the evaluation, each model is tested three times, with the average value taken as the final result. For evaluation purposes, all models are adjusted to a temperature of 0.2. The vLLM(Kwon et al., 2023) framework is employed for inference, with the maximum length restricted to 4096 tokens.

Table 15: **Evaluations of VLMs across different driving tasks on DriveBench** Each column shows the GPT score (↑) for the clean version of each task. We highlight the best-performing open-source model for each task.

| Method | Size | 👾 Perception | 🧪 Prediction | 🏛️ Planning | 🦺 Behavior |
|---|---|---|---|---|---|
| InternVL3 (Zhu et al., 2025) | 8B | 33.21 | 41.21 | 70.97 | 51.47 |
| DriveLMM-o1 (Ishaq et al., 2025) | 8B | 34.31 | 43.95 | 73.7 | 47.62 |
| DriveRX | 8B | 32.88 | 52.20 | 78.63 | 62.02 |

### O.3  LATENCY OF LLM-BASED REWARD SCORING

During the GRPO training process, we deployed Qwen2.5-72B-Instruct on 8×A100 using the vLLM framework for LLM-based reward scoring, achieving a throughput of 1,024 samples in 5 minutes (0.29 seconds per sample with batch size 128).

## P  COMPARISON WITH RECENT WORK USING DRIVEBENCH EVALUATION

DriveLMM-o1(Ishaq et al., 2025) is a recent work closely related to ours, also targeting multi-task visual-language reasoning in autonomous driving. It constructs a step-by-step reasoning dataset covering perception, prediction, and planning tasks, and fine-tunes an InternVL2.5-8B(Chen et al., 2024b) model using LoRA to improve logical coherence and interpretability in decision making.

Compared to DriveLMM-o1, our method introduces several key extensions: (1) We incorporate the behavior decision-making task, thereby covering the full reasoning chain in autonomous driving. (2) Instead of relying on supervised fine-tuning, we adopt AutoDriveRL, a joint reinforcement learning framework that leverages task-specific reward models to optimize the four semantically distinct subtasks in a coordinated manner. (3) While interpretability remains important, our primary objective is to enhance robustness and generalization in complex driving scenarios through structured, reward-guided reasoning.

Since both models are trained on nuScenes-derived data, we conduct a comparison on the DriveBench evaluation set, which also builds upon nuScenes. As shown in Table 15, DriveRX significantly outperforms DriveLMM-o1 on the prediction, planning, and behavior tasks, demonstrating the effectiveness of AutoDriveRL in learning structured, cross-task reasoning. Notably, this result also supports our earlier finding (Section F) that a stronger vision encoder contributes substantially to perception performance. While DriveRX adopts a CLIP-based encoder with efficient visual-

Table 16: OOD Evaluation on DriveAction Benchmark. The results report Action Accuracy.

| Model | V-L-A | V-A | L-A | A |
|---|---|---|---|---|
| GPT-4o (OpenAI, 2024a) | 88.84 | 84.72 | 86.52 | 81.01 |
| o3 (OpenAI, 2025) | 92.19 | 86.61 | 88.66 | 82.23 |
| Claude 3.7 Sonnet (Claude, 2025) | 86.31 | 80.80 | 82.56 | 80.67 |
| Align-DS-V (PKU-Alignment, 2025) | 78.58 | 77.32 | 77.74 | 76.67 |
| **DriveRX** | **88.09** | **86.56** | **87.17** | **86.18** |

language alignment, DriveLMM-o1 leverages the InternVL2.5 encoder, which offers higher capacity for fine-grained visual understanding. As a result, DriveLMM-o1 retains an advantage in the perception task, highlighting the importance of high-resolution visual representations for low-level scene comprehension.

## Q   LLM USAGE

In this work, large language models (LLMs) were employed as general-purpose assistive tools throughout the research process. Specifically, LLMs were used for text refinement and typo correction to improve the clarity and readability of the manuscript. These contributions were crucial in enhancing the overall quality of the text.

Additionally, LLMs played a significant role in the experimental setup. In our experiments, LLMs were utilized as reward models (RMs) to optimize the performance of reinforcement learning (RL) agents. The LLM-based reward models were trained and evaluated on specific tasks, providing valuable feedback and improving the agent's decision-making process. This approach was critical in fine-tuning the behavior of the models and achieving desired outcomes in various tasks.

All content generated or refined by LLMs was reviewed and validated by the authors to ensure scientific accuracy and integrity.

## R   OOD EVALUATION ON DRIVEACTION

To further assess the out-of-distribution (OOD) performance of DriveRX, we conducted evaluations on the **DriveAction** (Hao et al., 2025) benchmark, whose data are sourced from real-world mass-production vehicles with strict manual curation, making it a challenging OOD proxy distinct from DriveRX's training data.

We employed the official evaluation settings of DriveAction, which utilize an action-rooted tree structure to assess decision-making. Specifically, the benchmark defines four evaluation modes based on the information provided in the prompt to guide the action prediction. First, the **Full Pipeline Mode (V-L-A)** provides the model with ground-truth QA pairs from both upstream Vision (V) tasks (e.g., traffic light detection) and Language (L) tasks (e.g., navigation following) as textual context, evaluating action performance under fully informed conditions. Second, the **Vision/Language-Only Modes (V-A / L-A)** inject QA pairs from only one modality into the input to evaluate reliance on specific information. Finally, the **Uninformed Mode (A)** provides no upstream QA pairs, requiring the model to generate the final action decision relying purely on raw sensor inputs and internal reasoning without high-level external guidance. This mode best reflects the capability of end-to-end autonomous driving systems. We compared **DriveRX** against baseline models and state-of-the-art (SOTA) VLMs across these metrics, with results presented in Table 16.

The results demonstrate strong OOD robustness. DriveRX significantly outperforms the base model (Align-DS-V) across all metrics on this unseen dataset. Notably, it achieves performance on par with SOTA closed-source models (GPT-4o, o3) and even surpasses most of them on the Action (A) metric (86.18 vs. 81.01/80.67), validating its capability to generalize to complex, real-world driving scenarios.

