# OpenReview forum: "DriveRX: A Vision-Language Reasoning Model for Cross-Task Autonomous Driving"
_ICLR.cc/2026/Conference — Submitted to ICLR 2026_

### Official Review · Reviewer_Xhmu · 2025-10-24

**Soundness:** 3
**Presentation:** 3
**Contribution:** 2
**Rating:** 2
**Confidence:** 3

**Summary:**

This paper presents DriveRX, a vision-language-action model designed to enhance reasoning and decision-making for autonomous driving. The authors aim to leverage language-based reasoning to handle complex or ambiguous driving cases and show some performance improvements over generalist vision-language models. The paper also includes an evaluation on a trajectory prediction benchmark to demonstrate the model’s potential for better planning.

**Strengths:**

The paper shows a clear pipeline for data construction and training framework of VLM training, and it shows some improvement over general-purpose models like Qwen and other large VLMs, which suggests that domain-specific fine-tuning helps the model better understand driving scenes.

**Weaknesses:**

I have some concerns about the motivation and experimental setup. Most recent works that use VLMs for driving focus on long-tail reasoning or trajectory prediction, and they evaluate on clear benchmarks with established baselines. This paper shows results in Table 3, but the setup isn’t well explained, so it’s hard to understand what the numbers actually mean or how fair the comparison is.

The model is not compared against any strong end-to-end driving systems, such as UniAD or VAD, which are the standard baselines for trajectory prediction.

It’s also missing comparisons with other VLM-based methods, such as DriveVLM, OmniDrive, or ORION. In fact, ORION is a very close baseline, and based on the results here, this proposed method doesn’t outperform it.

Overall, the improvement seems marginal, and the paper doesn’t convincingly show that this method leads to any meaningful gains in driving performance. The motivation would be stronger if the authors could clearly demonstrate where this approach provides a unique advantage over existing systems.

**Questions:**

Can the authors clarify the evaluation setup for Table 3? How are the metrics calculated?

Why are the results not compared with other domain-specific models (e.g., UniAD, VAD, DriveVLM, OmniDrive, ORION)?

What's the main usecase for this VLM? Is it aiming for improving existing self driving system?

---

> ### Author Response · Authors · 2025-11-20
>
> We sincerely thank the reviewer for the detailed and constructive comments, especially regarding the motivation and application scope of our work. Below we provide point-by-point responses to address all concerns.
>
> ---
> ### **Q1 Clarify of the evaluation setup for Table 3**
>
> The evaluation presented in Table 3 focuses on the open-loop testing of the DriveRX-Agent model on the nuScenes dataset [1]. Building upon DriveRX, the agent extends the model's vocabulary by discretizing trajectory coordinates (covering a future horizon of 6 seconds) into 512 bins to define distinct trajectory tokens. The model is fine-tuned using reasoning data distilled from DriveRX combined with discretized ground-truth trajectory data. Performance is assessed using two metrics: ADE (Average Displacement Error) and Col. (Collision Rate). ADE measures prediction accuracy by calculating the average distance error between the predicted and ground-truth trajectories, while Col. quantifies the frequency with which predicted trajectories result in collisions with other objects in the scene. For both metrics, lower values indicate superior prediction performance and enhanced safety.
>
> All evaluations are conducted using the same settings as those in DriveLM [2] to ensure consistency and fairness.
>
> [1] C. Caesar et al., *nuScenes: A Multimodal Dataset for Autonomous Driving*, CVPR, 2020.
>
> [2] Sima C. et al., DriveLM: Driving with Graph Visual Question Answering, ECCV 2024.
>
> ---
>
> ### **Q2 Other domain-specific baseline models**
>
> As shown in **Table 4**, we include comparisons against **UniAD, VAD, and ORION**, which represent strong baselines across both planning-oriented and VLM-enhanced driving systems.
>
> As discussed in **Section 6**, our goal is **not to replace specialized planners** but to demonstrate that a reasoning-enhanced VLM can serve as a unified backbone for both high-level decision making and low-level control. DriveRX achieves competitive open-loop and closed-loop performance, and although it trails the SOTA on certain metrics, the results consistently show that **reasoning-based distillation improves planning quality and action generation**. These findings confirm the feasibility and practical value of adopting a unified VLM-driven framework for multi-stage autonomous driving.
>
> ---
>
> ### **Q3 Motivation and Application**
>
> We would like to further clarify the motivation and intended application scenarios of DriveRX.
>
> DriveRX is **not** designed to function as a real-time low-level planner or vehicle-control backbone. Instead, it targets **cross-task vision–language reasoning**, focusing on high-level semantic understanding, intention prediction, and interpretable decision reasoning. Thus, DriveRX does not aim at low-latency control.
>
> Our results on DriveBench and DriveLM-Hard show that DriveRX generates stable and interpretable reasoning chains under complex and long-tail cases, supporting its role as a backbone for high-level semantic reasoning and consistent cross-task decision making. These chains also provide reliable stage-wise semantics, which are useful for annotation assistance, hard-case diagnosis, and reasoning-quality inspection.
>
> In addition, the high-level reasoning outputs of DriveRX can serve as **teacher signals or reward feedback** for downstream planning or control models. For this reason, Section 6 further discusses two applications—DriveRX-Agent and DriveRX-VLA—which validate the effectiveness of such “reasoning-enhanced” supervision.
>
> In summary, the purpose of this work is **not** to report gains in real-time low-level driving performance, but to demonstrate the value of a reasoning-centered VLM backbone that unifies semantic perception, prediction, planning, and behavior abstraction, and that can benefit downstream autonomous driving modules. While DriveRX’s structured data and reasoning outputs can indeed be used to further improve low-level autonomous driving models through targeted training, this direction is beyond the scope of the present work and we leave it for future research.
>
> We will further clarify these points in **Section 1 (Introduction)** and **Appendix A (Broader Impact)** of the revised manuscript.

---

> ### Author Response · Authors · 2025-11-27
>
> Dear Reviewer **Xhmu**,
>
> Hello! Thank you very much for your thoughtful review and for sharing your concerns regarding our submission. We truly appreciate the time and effort you have invested in evaluating our work.
>
> We have carefully addressed each of the issues you raised in our rebuttal and have also updated the revised version of the paper (PDF) accordingly. Your comments were extremely helpful in improving the clarity and quality of our work.
>
> We sincerely invite you to take another look at our updated responses and revised manuscript. If our clarifications have resolved your concerns, we would be deeply grateful if you could kindly reconsider your evaluation. Should any questions or doubts remain, we would be more than happy to provide further explanation or make additional improvements.
>
> Thank you again for your valuable feedback and consideration.
>
> Sincerely,
> The authors

---

### Official Review · Reviewer_3U7X · 2025-10-30

**Soundness:** 2
**Presentation:** 3
**Contribution:** 3
**Rating:** 6
**Confidence:** 5

**Summary:**

This paper presents DriveRX, a vision-language reasoning model for autonomous driving trained via the AutoDriveRL reinforcement learning framework.
AutoDriveRL decomposes driving into perception, prediction, planning, and behavior tasks, each optimized with task-specific reward models.
Using structured reasoning and joint RL training, DriveRX achieves strong generalization and robustness on DriveBench and DriveLM-Hard benchmarks, outperforming GPT-4o in behavior reasoning.
The model also supports downstream trajectory prediction and action generation through reasoning-based distillation.

**Strengths:**

+ Proposes a unified RL framework (AutoDriveRL) enabling interpretable, multi-stage reasoning across driving subtasks.

+ Demonstrates state-of-the-art performance and robustness, surpassing larger models under both clean and corrupted conditions.

+ Extends practical impact by showing reasoning-enhanced transferability to trajectory and control tasks.

**Weaknesses:**

**For the image input**: In the structured reasoning process you designed, including prediction and planning, you only feed the model multi-view images, which makes it very hard to ensure the model can capture temporal information like speed — this doesn’t really make sense. Although some other VLAs also only use images, they at least provide historical ego‐states to ensure the ego car’s temporal information.

**Application**: The authors design a very complex reasoning process; I am quite puzzled about what scenario the authors expect their work to be applied in. If it’s just high-level prediction, then they have not opened up a large gap compared with current open-sourced VLMs such as InternVL3-8B, Qwen3-VL-8B, etc. (And I must emphasize: the authors test on DriveBench but that is not truly OOD because both DriveRX and DriveBench data are constructed based on DriveLM.) I am not sure whether DriveRX has better performance than the open-sourced VLMs in other domains of driving scenarios (e.g., DriveAction).

**Latency**: Because I am confused about the application of DriveRX. If it is used for low-level planning, then latency is a big challenge; and based on Table 3 results, its performance gap with existing VLAs model is significant (e.g., Auto-VLA).

**Questions:**

+ Could the authors clarify the time latency when using an LLM for scoring during training?

+ For the prediction and planning tasks, did the authors explore using a rule-based reward model? If so, how does it compare (advantages/disadvantages) to the LLM-based reward?

+ Could the authors provide results on the AlphaDrive benchmark for planning, so we can assess true out-of-distribution (OOD) performance?

+ Based on Table 8, I don’t understand why the gap between SFT (supervised fine-tuning) and RL in the behavior task is so large—this phenomenon isn’t observed in other related work. Could the authors explain?

+ **Most importantly, I’d like the authors to clearly articulate the intended application scenarios for DriveRX:**

(i) As a strong backbone for low-level planning tasks?

(ii) As a data annotator for long-tail driving scenarios?

(iii) As a reward model to improve other VLA/VLMs’ low-level planning performance?

---

> ### Author Response · Authors · 2025-11-20
>
> We sincerely thank the reviewer for the careful evaluation and constructive feedback. Your insightful suggestions will help us further improve the paper, and we address each point in detail below.
>
> ### **Q1 Latency of LLM-Based Reward Scoring**
>
> In our setting, the overhead is manageable due to efficient inference frameworks such as vLLM [1]. Based on our training logs, deploying Qwen2.5-72B-Instruct on 8×A100 yields a throughput of 1,024 scoring in approximately 5 minutes, corresponding to 0.29 seconds per sample with batch size 128. Each RL step therefore takes about 15 minutes, which is fully acceptable in our training pipeline. We will add these latency statistics to the revised version for clarity.
>
> We note that using an LLM judge for reward scoring has become a standard practice in recent RL-for-LLM works. For instance, the Qwen3 Technical Report also uses Qwen2.5-72B-Instruct to score model outputs against reference answers. This approach provides flexible, task-aware evaluation and avoids the brittleness of purely rule-based reward functions.
>
> We will integrate this clarification into the implementation details in Appendix O in the revised manuscript.
>
> [1] Kwon, Woosuk, et al. “Efficient memory management for large language model serving with PagedAttention.” SOSP 2023.
>
> [2] Yang A, Li A, Yang B, et al. Qwen3 technical report[J]. arXiv preprint arXiv:2505.09388, 2025.
>
> ---
>
> ### **Q2 Use of Rule-Based Reward**
>
> We explored rule-based and hybrid reward models, and we report the corresponding experiments in **Appendix J**.
>
> Traditional rule-based metrics are not well aligned with the semantic nature of these tasks. For instance, when the predicted list of agents differs in order or cardinality from the reference, IoU-based set matching quickly becomes unreliable; similarly, n-gram text metrics (e.g., BLEU/ROUGE) penalize valid paraphrases that convey the correct semantics. In our experiments, RL training driven solely by rule-based rewards was highly unstable and frequently collapsed.
>
> To address this, we implemented a **hybrid reward scheme** that combines rule-based scores with LLM-based scores (details in Appendix J). Although the hybrid approach provides more fine-grained signals, it consistently introduced **significant optimization instability**: reward curves exhibited sharp oscillations and lacked a clear convergence trend. As shown in **Fig. 11**, even the best-performing hybrid runs fluctuate heavily, in contrast to the smooth and monotonic improvements achieved with pure GPT-based rewards.
>
> We attribute this instability to the inconsistent scales of different rule-based metrics, conflicting optimization objectives, and the sensitivity of structural matchers to small output variations.
>
> Therefore, we ultimately adopted the pure LLM-based reward, as it offered stable RL training dynamics and more reliable assessment of model responses.
>
> ---
>
> ### **Q3 Evaluation on OOD Distributions (DriveAction).**
>
> Since the AlphaDrive benchmark is not publicly accessible, we employed **DriveAction** to rigorously evaluate OOD performance. Sourced from real-world mass-production vehicles with strict manual curation, DriveAction serves as a challenging OOD proxy distinct from our training data (DriveLM). Utilizing the official evaluation scripts, we compare DriveRX against the base model and SOTA VLM agents:
>
> | **Model** | **V-L-A** | **V-A** | **L-A** | **A** |
> | --- | --- | --- | --- | --- |
> | GPT-4o | 88.84 | 84.72 | 86.52 | 81.01 |
> | o3 | 92.19 | 86.61 | 88.66 | 82.23 |
> | Claude 3.7 Sonnet | 86.31 | 80.80 | 82.56 | 80.67 |
> | Align-DS-V | 78.58 | 77.32 | 77.74 | 76.67 |
> | **DriveRX** | **88.09** | **86.56** | **87.17** | **86.18** |
>
> The results demonstrate **strong OOD robustness**. DriveRX significantly outperforms the base model (Align-DS-V) across all metrics on this unseen dataset. Notably, it achieves performance on par with SOTA closed-source models (GPT-4o, o3) and even **surpasses most of them on the Action (A) metric** (86.18 vs. 81.01/80.67), validating its capability to generalize to complex, real-world driving scenarios.
>
> We will include this OOD evaluation in the appendix of the revised manuscript.
>
> ---

---

> > ### Author Response · Authors · 2025-11-20
> >
> > ### **Q4 The gap between SFT and RL**
> >
> > The significant gap arises because driving behavior requires **long-chain reasoning** (Perception→Prediction→Planning→Behavior), rather than simple pattern recognition.
> >
> > **(1) SFT Limitations:** SFT only mimics static, non-reasoning annotations. It struggles to internalize the complex, multi-step logic required for high-level decision-making. (2) **RL (GRPO) Advantages:** Through Group Relative Policy Optimization (GRPO), the model explicitly explores multiple reasoning paths to identify optimal strategies, effectively learning "how to think" rather than just "what to say." (3) **Task Complexity:** Unlike simple VQA tasks (e.g., in *R1-Onevision*) where the SFT-RL gap is minimal, autonomous driving involves complex logic and safety constraints. The large gap confirms that behavior generation is a reasoning-heavy task that benefits disproportionately from the exploration and self-correction capabilities of RL.
> >
> > We will include this analysis in **Appendix K** of the revised paper.
> >
> > ---
> >
> > ### **Q5 Clarify the intended application scenarios for DriveRX**
> >
> > Thank you for the thoughtful comments. The intended application scope of DriveRX indeed requires clarification. First, DriveRX is **not** designed to serve as a real-time low-level planning or vehicle control backbone. Instead, it focuses on cross-task vision–language reasoning, including high-level semantic understanding, intention prediction, and interpretable decision reasoning, and therefore does not target low-latency control.
> >
> > Our main experimental results on DriveBench and DriveLM-Hard demonstrate that DriveRX provides stable and interpretable reasoning chains under complex and long-tail scenarios. Thus, its primary application is as a backbone for high-level semantic reasoning and consistent cross-task decision making, rather than for low-level control. Owing to its structured Perception–Prediction–Planning–Behavior reasoning chain, DriveRX also produces reliable stage-wise semantic analyses in difficult cases, making it useful for assisting annotation, hard-case diagnosis, and reasoning-quality inspection.
> >
> > In addition, the high-level reasoning outputs of DriveRX can serve as teacher signals or reward feedback for downstream planning/control models. The experiments on DriveRX-Agent and DriveRX-VLA further validate the effectiveness of this “reasoning-enhanced” supervision.
> >
> > In summary, DriveRX is positioned as a cross-task high-level semantic reasoning model that improves understanding and decision consistency in complex scenarios, and provides high-quality supervisory signals for annotation and downstream models. It is **not** intended to replace real-time low-level planning or control modules.
> >
> > We will revise **Section 1 (Introduction)** and **Section 6 (Applications)** in the revised manuscript to clearly clarify these intended application scenarios.

---

> ### Author Response · Authors · 2025-11-27
>
> Dear Reviewer **3U7X**,
>
> Hello! Thank you once again for your positive recognition of our work and for your constructive, insightful comments. We have carefully addressed your feedback and have also updated the revised version of our paper (PDF). We sincerely invite you to take another look and reassess our submission in light of these clarifications.
>
> If our responses have adequately resolved your concerns, we would be truly grateful if you could kindly reconsider the score. Should you have any remaining questions or further suggestions, we would be more than happy to continue the discussion and make additional improvements.
>
> We truly appreciate your time, effort, and thoughtful consideration.
>
> Sincerely,
> The authors

---

> ### Comment · Reviewer_3U7X · 2025-11-27
>
> Thank you for the very detailed response and for updating the manuscript. Your clarifications on latency, reward design, OOD evaluation, the SFT–RL gap, and application scope have addressed most of my earlier concerns.
>
> I still have one remaining question about the image input / temporal limitation, which was not fully addressed. DriveAction provides 3 past frames, while DriveRX seems to operate on single images without explicit temporal modeling. How exactly do you test on DriveAction—do you only feed the last frame, or aggregate the 3 frames somehow? If no temporal information is used, it is unclear how the model can reliably reason about other vehicles’ future behavior, given that ego behavior depends heavily on surrounding agents’ motion.
>
> Finally, from a practical perspective: if a practitioner is already willing to rely on a strong closed-source model, why should they use DriveRX as an annotator/teacher instead of directly calling GPT-4o (or similar) for labels?
>
> BTW, I think the demonstrations you have shown are too simple. Figures 11, 12, and 13 show that there were no moving vehicles in the neighbourhood of the ego vehicle.

---

> > ### Author Response · Authors · 2025-11-28
> >
> > Thank you for your thoughtful follow-up comments and for carefully reviewing our revised manuscript. We appreciate the opportunity to further clarify the remaining concerns. Below we provide answers to each of your questions.
> >
> > ### **Q: Evaluation on DriveAction**
> >
> > DriveRX is indeed trained using six camera views from a single timestamp. During evaluation on DriveAction, the model does **not** rely on a single static frame. Instead, we provide the images at timestamps *t-2*, *t-1*, and *t* sequentially as inputs, together with a text prompt explicitly indicating that they correspond to the three consecutive frames preceding the decision moment. This allows the model to incorporate short-term temporal cues at test time, even though the training data itself does not contain explicit temporal modeling.
> >
> > We will add this clarification and the prompt to Appendix R in the revised manuscript.
> >
> > ---
> >
> > ### **Q: Why not use GPT-4o?**
> >
> > There are three main considerations.
> >
> > **First, cost and scalability:** our 8B model can be deployed on a single GPU, is open-source and free, and therefore supports large-scale data labeling. In contrast, GPT-4o is a paid, closed-source model.
> >
> > **Second, data privacy:** DriveRX supports fully local deployment, which is crucial because autonomous-driving data is limited and highly sensitive. Local deployment ensures that no private data ever leaves the system, whereas GPT-4o requires sending data to an external commercial platform.
> >
> > **Third, engineering practicality:** integrating DriveRX into RL training (e.g., reward labeling) only requires a single locally running GPU. Using GPT-4o would rely on external API calls and internet connectivity, introducing latency and instability that make RL experiments significantly harder to run reliably.
> >
> > ---
> >
> > We will include this discussion in the revised manuscript and add additional case studies involving surrounding vehicles in motion in the updated PDF. Once again, we sincerely thank you for your helpful comments.

---

### Official Review · Reviewer_6G7c · 2025-10-31

**Soundness:** 3
**Presentation:** 3
**Contribution:** 2
**Rating:** 6
**Confidence:** 4

**Summary:**

This paper proposes AutoDriveRL, a reinforcement learning (RL) framework for vision-language models (VLMs) applied to autonomous driving, decomposing driving into four interpretable sub-tasks: Perception, Prediction, Planning, and Behavior. Each sub-task is framed as a VQA problem with a task-specific reward model. Based on this framework, the authors train DriveRX, a cross-task reasoning VLM that aims to unify multi-stage decision-making. Experiments on DriveBench and DriveLM-Hard show that DriveRX outperforms GPT-4o and other baselines in the Behavior task and maintains robustness under visual corruptions. They further extend DriveRX to (1) DriveRX-Agent for trajectory generation and (2) DriveRX-VLA for closed-loop control via reasoning-based distillation.

**Strengths:**

This study introduces a reinforcement-learning–driven, multi-task vision-language reasoning framework designed to enhance the coordination between perception, reasoning, and action in autonomous driving. By attaching task-specific reward models to each stage of the reasoning process, the approach provides finer-grained supervision, improving interpretability and stability.

It organizes the problem into four subtasks that reflect the cognitive pipeline of autonomous driving—from perception to behavior. A Visual Question Answering (VQA)–style data formulation is used to unify the data structure, supporting cross-task consistency and a cleaner, more integrated learning signal.

Empirical results show that DriveRX achieves 62.02 on the Behavior task, outpacing GPT-4o (55.04) and the strongest open-source vision-language models. The method also demonstrates robust performance under visual corruption, underscoring its resilience. Baseline comparisons cover a broad spectrum, including commercial models (e.g., GPT-4o), generalist models (LLaVA, Qwen2.5-VL), reasoning-oriented models (MM-Eureka, R1-OneVision), and domain-specific systems (DriveLM, Dolphins), all evaluated under consistent settings.

Beyond core reasoning, the approach extends to trajectory prediction and control through DriveRX-Agent and DriveRX-VLA. These extensions illustrate how high-level reasoning can be bridged with low-level action generation, pointing toward a unified driving agent that blends perception, reasoning, and control in a single framework.

**Weaknesses:**

1. Ambiguous Reinforcement Learning Details
Equation (2) defines GRPO, but the intra-group advantage term (Eq. 3) is shared across tokens, not time-dependent, this simplifies training but may degrade credit assignment.
No mention of reward normalization schedule, policy rollout horizon, or sampling temperature, which are essential for RL reproducibility.
The rule-based and LLM-based reward models are described qualitatively but lack explicit scoring functions, thresholds, or examples (Appendix M.1 is referenced but not shown in main text).

2. Missing Quantitative Ablations
The framework introduces multiple key design choices (reward composition, GRPO vs PPO, per-task weighting, data filtering), yet no ablation results are provided in the main paper.
For example, how much does AutoDriveRL contribute compared to supervised fine-tuning on the same data? The gain from GRPO remains unquantified.

3. Unclear Task Coupling
The paper claims cross-task reasoning, but tasks are trained independently with shared parameters, without conditioning on previous task outputs (explicitly stated in §3.1).
This design limits true reasoning chaining; the reasoning chain is only semantic, not computationally causal. Thus, DriveRX may not genuinely achieve process-level reasoning across tasks.

4. Evaluation Ambiguities
The GPT score metric is used exclusively, judged by GPT-3.5-Turbo. This raises reliability and circularity issues — the model is trained with LLM-based rewards and then evaluated by another LLM judge.
There is no mention of inter-rater agreement, or calibration with human labels (Appendix I claims alignment but gives no correlation coefficients in the main text).
Metrics like ADE/Col. in Table 3 and Driving Score in Table 4 are reported but lack variance or standard deviation, leaving statistical significance unclear.

5. Distillation Pipeline Unclear
In §6, reasoning data from DriveRX is distilled into DriveRX-VLA, but the exact mapping from language to action tokens and supervision loss are unspecified.In §6, reasoning data from DriveRX is distilled into DriveRX-VLA, but the exact mapping from language to action tokens and supervision loss are unspecified.
It’s not clear whether the VLA model receives image input, text prompts, or both during inference — this affects the claimed “unified backbone” property.

6. Interpretability and Example Limitation
Only one qualitative case (Figure 3) is provided. No multi-turn reasoning trace or counterfactual analysis is shown, weakening claims of interpretability.

**Questions:**

1. Ambiguous Reinforcement Learning Details
Equation (2) defines GRPO, but the intra-group advantage term (Eq. 3) is shared across tokens, not time-dependent, this simplifies training but may degrade credit assignment.
No mention of reward normalization schedule, policy rollout horizon, or sampling temperature, which are essential for RL reproducibility.
The rule-based and LLM-based reward models are described qualitatively but lack explicit scoring functions, thresholds, or examples (Appendix M.1 is referenced but not shown in main text).

2. Missing Quantitative Ablations
The framework introduces multiple key design choices (reward composition, GRPO vs PPO, per-task weighting, data filtering), yet no ablation results are provided in the main paper.
For example, how much does AutoDriveRL contribute compared to supervised fine-tuning on the same data? The gain from GRPO remains unquantified.

3. Unclear Task Coupling
The paper claims cross-task reasoning, but tasks are trained independently with shared parameters, without conditioning on previous task outputs (explicitly stated in §3.1).
This design limits true reasoning chaining; the reasoning chain is only semantic, not computationally causal. Thus, DriveRX may not genuinely achieve process-level reasoning across tasks.

4. Evaluation Ambiguities
The GPT score metric is used exclusively, judged by GPT-3.5-Turbo. This raises reliability and circularity issues — the model is trained with LLM-based rewards and then evaluated by another LLM judge.
There is no mention of inter-rater agreement, or calibration with human labels (Appendix I claims alignment but gives no correlation coefficients in the main text).
Metrics like ADE/Col. in Table 3 and Driving Score in Table 4 are reported but lack variance or standard deviation, leaving statistical significance unclear.

5. Distillation Pipeline Unclear
In §6, reasoning data from DriveRX is distilled into DriveRX-VLA, but the exact mapping from language to action tokens and supervision loss are unspecified.In §6, reasoning data from DriveRX is distilled into DriveRX-VLA, but the exact mapping from language to action tokens and supervision loss are unspecified.
It’s not clear whether the VLA model receives image input, text prompts, or both during inference — this affects the claimed “unified backbone” property.

6. Interpretability and Example Limitation
Only one qualitative case (Figure 3) is provided. No multi-turn reasoning trace or counterfactual analysis is shown, weakening claims of interpretability.

---

> ### Author Response · Authors · 2025-11-20
>
> Thank you for the detailed comments.Below we provide point-by-point answers to your questions.
>
> ### **Q1(a): Intra-group advantage shared across tokens**
>
> First, our intent is not to develop a new RL method, but to employ a well-established RL paradigm that enables the model to conduct rollouts and be optimized according to task-specific rewards in autonomous driving reasoning.
>
> Accordingly, we adopt the standard GRPO algorithm from [1]. In Eq. 3, all tokens within a sampled trajectory indeed share the same reward. This behavior is fully aligned with the reward structure and matches the default design of mainstream LLM-RL methods (GRPO [1]/ DAPO [2]/ GSPO [3]). Specifically:
>
> **(1) Our driving-reasoning tasks inherently produce sequence-level rewards.**
>
> For each of the four driving subtasks, every candidate response receives one holistic quality score (correctness, behavioral understanding, contextual relevance, reasoning clarity, etc.).
>
> Such a reward cannot be decomposed at the token level.
>
> The GRPO paper explicitly states that the reward is sequence-level:
>
> > “the reward model usually only assigns a score at the last token.” (DeepSeekMath, Sec. 4.1.1)
> >
>
> Therefore, assigning the same advantage to all tokens is **the intended design of GRPO** and directly matches our task formulation.
>
> **(2) Sequence-level (scalar) advantages are standard in LLM-based RL.**
>
> Whenever the reward is defined at the sequence level, algorithms such as GRPO, DAPO, and GSPO all operate with a single scalar advantage per generated output.
>
> Thus, our use of a sequence-level advantage is consistent with widely adopted practice in LLM-RL.
>
> **(3) Empirically, our RL training is stable and effective.**
>
> As shown in Fig. 2, our training curves exhibit smooth optimization, and the reward consistently increases across all four tasks. This empirical behavior indicates that using a shared advantage **does not harm credit assignment in practice**, and the learning dynamics remain stable and well-behaved.
>
> [1] Shao Z. et al. DeepSeekMath: Mathematical Reasoning in Open LLMs. arXiv:2402.03300, 2024.
>
> [2] Yu Q. et al. DAPO: An Open-Source LLM RL System at Scale. arXiv:2503.14476, 2025.
>
> [3] Zheng C. et al. Group Sequence Policy Optimization. arXiv:2507.18071, 2025.
>
> ---
>
> ### **Q1(b): Missing RL hyperparameters**
>
> The reward normalization schedule is already specified in Eq. 3, where we apply group-level Z-score normalization.
>
> For each task, the reward model described in Appendix M.1 assigns a scalar score in the range 0–100 using task-specific LLM prompts, and the final advantage is computed using the Eq. 3 in Section3.4.
>
> Regarding additional RL details, the policy rollout horizon, sampling temperature, and other training hyperparameters are fully listed in Appendix O.2 (Hyperparameter Settings).
>
> We will make this cross-reference clearer in the revised version.
>
> ---
>
> ### **Q1(c): Reward model scoring**
>
> Appendix M.1 (Fig. 17–21) provides the prompts used for LLM-based reward scoring, along with detailed scoring criteria and numeric guidelines for each task.
> Appendix I reports the consistency between different reward models and human annotations, showing a high alignment between LLM scores and human ground truth (Pearson r > 0.83).
> Appendix J compares the LLM-judge reward with combinations of LLM scores and rule-based scores across tasks, and we find that the LLM score is more reliable and yields more stable training curves (Fig. 11).
>
> ---

---

> > ### Author Response · Authors · 2025-11-20
> >
> > ### **Q2 Quantitative Ablations**
> >
> > We provide quantitative ablations in the appendix. Appendix J compares the LLM-judge reward with mixed rule-based schemes and shows that LLM scores yield more reliable evaluation and significantly more stable training curves (Fig. 11). Appendix C reports per-task sample statistics and data-filtering results (Fig. 4–6), with the full filtering strategy described using Eq. 1. In Appendix K (Table 8), we further compare DriveRX-Joint with DriveRX-SFT trained on the same data, showing clear improvements from our RL training. Since our task cannot support a stable critic, we adopt standard GRPO and therefore do not include a PPO comparison.
> >
> > We additionally include a controlled ablation to isolate the effect of our data-filtering strategy.
> >
> > Specifically, we randomly sample from the *raw* dataset the same number of training instances as in the filtered dataset, keep all training configurations identical, and evaluate on DriveLM-Hard.
> >
> > |  | Perception | Prediction | Planning | Behavior |
> > | --- | --- | --- | --- | --- |
> > | Align-DS-V(base) | 29.45 | 52.90 | 29.15 | 32.48 |
> > | DriveRX w/ Data Filtering | 34.83 | 71.84 | 51.42 | 36.82 |
> > | DriveRX w/o Data Filtering | 30.29 | 51.82 | 48.23 | 33.07 |
> >
> > We will include the full training curves in the revised appendix. Empirically, the model trained on the filtered corpus shows (i) **much faster improvement on the validation set from the very first steps**, and (ii) **consistently higher performance** compared with training on the unfiltered data. On DriveLM-Hard, the filtered-data model outperforms the unfiltered baseline across multiple tasks, demonstrating that our filtering pipeline provides cleaner supervision signals and materially improves RL training stability and final performance.
> >
> > ---
> >
> > ### **Q3 Cross-task reasoning**
> >
> > Our formulation does not use an explicit step-by-step pipeline that passes textual outputs between tasks. Instead, DriveRX learns **cross-task structural dependencies** through shared parameters and a unified latent representation. Although the four tasks are trained on separate samples, they update the same model and jointly shape the underlying representation space.
> >
> > This cross-task dependency is empirically supported. In the DriveLM-Hard ablations (Appendix G, Table 6), **removing any single subtask leads to a notable drop in behavior performance**, suggesting that the model leverages complementary information across tasks rather than treating them as isolated semantic queries.
> >
> > Furthermore, our comparison with the Sequential Cascade baseline (Appendix G, Table 6) shows that explicitly chaining task outputs tends to propagate upstream errors and yields worse performance. This indicates that, in our setting, shared latent representations provide a more stable way to capture cross-task reasoning than explicit pipelined conditioning.
> >
> > ---
> >
> > ### **Q4 LLM Evaluation**
> >
> > Our GPT-based evaluation is not an open-ended subjective judgment. The judge model assesses consistency between each response and task-specific reference answers, following explicit rubrics for Perception, Prediction, Planning, and Behavior.
> >
> > To verify reliability, Appendix I reports a Pearson correlation of r = 0.83 between GPT-4o scores and human annotations collected from three independent annotators. We will move this result into the main text for clarity.
> >
> > Moreover, although both the reward model and the evaluation model rely on task-specific reference answers, they use different LLMs and independent scoring prompts, so the evaluation does not introduce circularity or self-reinforcing bias.
> >
> > Finally, as stated in Appendix O.2, evaluations are repeated three times, and we report the average to ensure stable results.
> >
> > ---
> >
> > ### **Q5 Distill pipeline**
> >
> > The distillation pipeline is detailed in Appendix H, where we describe how DriveRX’s reasoning traces are mapped into discrete action tokens for DriveRX-VLA training. During both VLA and Agent inference, the model always multimodal inputs — including RGB images and ego-state features — rather than text alone. This ensures that all modules operate under the same unified multimodal backbone.
> >
> > For supervision, DriveRX-VLA is trained using standard supervised fine-tuning (cross-entropy loss) on the distilled action sequences. No additional loss terms or special objectives are used. We will clarify these points explicitly in the Section 6.
> >
> > ---
> >
> > ### **Q6 Case Studies**
> >
> > Beyond the example in Figure 3, we provide comprehensive case studies in **Appendix L (Figures 12–16)**. These include detailed reasoning traces across all four tasks and comparisons between different model variants, clearly showing how the model performs step-by-step reasoning. We will add explicit pointers in the main text to ensure these analyses are easier to locate.
> >
> > ---
> >
> > Finally, we sincerely thank the reviewer for the detailed comments. We will incorporate your suggestions into the revised manuscript and update the PDF accordingly.

---

> ### Author Response · Authors · 2025-11-27
>
> Dear Reviewer **6G7c**,
>
> Thank you very much for your thoughtful comments and valuable feedback. We have carefully addressed each of your concerns in our rebuttal, and we trust you have had a chance to review our responses, and we have also updated our revised paper PDF accordingly.
>
> If our clarifications have addressed your concerns, we would greatly appreciate it if you could kindly consider updating your score. If there are any remaining questions or aspects of our work you would like to discuss further, we would be glad to engage in deeper discussion.
>
> Sincerely,
>
> The authors

---

### Official Review · Reviewer_7UfE · 2025-11-04

**Soundness:** 3
**Presentation:** 3
**Contribution:** 2
**Rating:** 4
**Confidence:** 4

**Summary:**

This paper introduces AutoDriveRL, a unified reinforcement learning framework that models autonomous driving as a structured reasoning process across perception, prediction, planning, and behavior. It trains a cross-task reasoning model, DriveRX, which outperforms GPT-4o in behavior reasoning and shows strong robustness in complex driving scenarios.

**Strengths:**

1. The formulation of each sub-task as a vision-language QA problem is interesting.

2. The experimental results are promising.

**Weaknesses:**

1. The introduction section feels somewhat wordy and difficult to follow. It’s not clearly structured around three key points: what common issues exist in current works, what direction and method are proposed, and how the proposed method is designed. These aspects should be laid out more clearly.

2. Additionally, the paper categorizes the autonomous driving pipeline into perception, prediction, planning, and behavior. Typically, autonomous driving is divided into three main tasks, perception, prediction, and planning, while behavior is often treated as part of prediction. What is the rationale for introducing behavior as a separate stage?

3. The description of the task-specific reward models lacks clarity. How the reinforcement signals are computed and whether they are consistent across tasks could be better elaborated.

4. It’s unclear how much each reasoning stage (perception, prediction, etc.) contributes to overall performance,  finer-grained ablation or visualization would improve interpretability.

**Questions:**

1. How robust is DriveRX to distribution shifts, e.g., unseen weather or sensor noise, beyond the “corrupted” conditions mentioned?

2. Are there any failure cases observed during deployment or simulation that reveal limits in the cross-task reasoning design?

---

> ### Author Response · Authors · 2025-11-20
>
> We thank the reviewer for the careful evaluation and constructive feedback. We address all comments and questions point-by-point below.
>
> ### **Q1 Clarity of the introduction**
>
> Thank you for the reviewer’s thoughtful feedback. We agree that the introduction can be better structured around three key elements: **(1) the common limitations in existing works, (2) the direction and methodology proposed in this paper, and (3) the design of our framework**. In the revised version, we will reorganize the introduction accordingly.
>
> **(1) Common limitations in current methods.**
>
> We will make the limitations more explicit by highlighting three shared issues in current VLM-for-driving approaches:
>
> - the **absence of intermediate reasoning chains**, which weakens interpretability and robustness in complex scenarios;
> - **fragmented task design** across perception, prediction, planning, and behavior, preventing consistent cross-task reasoning;
> - **reliance on static supervision**, which lacks adaptive feedback for improving reasoning quality.
>
> **(2) Our proposed direction and methodology.**
>
> We will emphasize the core idea of this paper: **formulating autonomous driving as a structured reasoning process across four tasks and introducing AutoDriveRL, an RL-based framework that provides stage-wise, task-specific feedback signals.**
>
> This clarification will make it clear how our direction directly addresses the above limitations and why structured reasoning combined with RL is essential for improving cross-task consistency and robustness.
>
> **(3) How our proposed framework is designed.**
>
> Finally, we will provide a concise overview of the key design components of AutoDriveRL:
>
> - decomposing autonomous driving into four semantically aligned subtasks (perception, prediction, planning, behavior);
> - designing **task-specific reward models** for each subtask;
> - jointly optimizing a unified VLM (DriveRX) through reinforcement learning across all tasks.
>
> This overview will prepare the reader for Section 3 and clarifies the contributions of the framework.
>
> We will revise the introduction according to this three-part structure, resulting in a clearer and more coherent presentation. These revisions will be incorporated into the updated PDF.
>
> ---
>
> ### **Q2 Why behavior is a separate stage**
>
> Although traditional autonomous driving pipelines typically follow a three-stage structure, we adopt the four-stage formulation (Perception–Prediction–Planning–Behavior) following prior work such as DriveLM [1]. Under this formulation:
>
> - **Behavior** is defined as a *discrete abstraction of the ego vehicle’s final driving decision* (e.g., “slowly go straight”, “prepare to turn right”).
>
> The Behavior stage serves as a lightweight and interpretable interface between high-level semantic reasoning (P1–P3) and the trajectory/action output. Separating it leads to a clearer semantic hierarchy: Prediction focuses on modeling others behavior, Planning formulates feasible, and Behavior summarizes the ego’s final decision.
>
> In addition, we further train a VLA (Vision–Language–Action) model, where this discrete behavior representation provides a stable intermediate supervision signal. Compared with directly regressing trajectories from the prediction stage, the Behavior abstraction enables more robust and reliable mapping from reasoning outputs to action tokens.
>
> Thus, the four-stage design is a deliberate modeling choice that aligns with established protocols while also improving semantic clarity, interpretability, and compatibility with downstream action learning. We will clarify this motivation in the revised manuscript.
>
> [1] Sima C. et al., DriveLM: Driving with Graph Visual Question Answering, ECCV 2024.
>
> ---
>
> ### **Q3 Task-Specific Reward**
>
> Each subtask has its own evaluation rubric (Appendix M.1), and the reward model assigns a 0–100 scalar score based on these criteria. This unified scale keeps the RL signals consistent and directly comparable across tasks (e.g., Perception vs. Planning). It also avoids issues caused by differences in score magnitude and ensures more stable joint optimization with the advantage function in Eq. 3. We will clarify this design in Section 3.4 of the revised paper.
>
> ---
>
> ### **Q4 Contribution of Individual Reasoning Stages**
>
> To evaluate the specific contribution of each reasoning stage, we have included fine-grained ablation studies in **Appendix G**. Specifically, we conducted experiments by selectively excluding the training data of individual stages to isolate their impact. As shown in **Table 6**, omitting any single stage leads to a noticeable decline in overall driving performance. This result empirically validates that every stage in the reasoning chain is indispensable and that they work synergistically to ensure the system's robustness.
>
> ---

---

> ### Author Response · Authors · 2025-11-20
>
> ### **Q5 OOD Evaluation**
>
> We clarify that DriveRX has been evaluated under both *corruption-based* and *real OOD* settings.
>
> **(1) Corruption robustness on DriveBench.**
>
> In **Table 1**, DriveRX is evaluated under the *corrupted* setting, which is **not a single perturbation**, but the **average score across 15 diverse visual corruption types**. These include weather-related and sensor-related degradations such as Brightness, Snow, Fog, Lens Obstacle, Water Splash, Camera Crash, Frame Lost, Motion Blur, Zoom Blur, Bit Error, Color Quantization, H.265 compression, etc. This setup stresses the model with heterogeneous input shifts rather than mild noise.
>
> **(2) Real-world OOD robustness on DriveAction.**
>
> To further assess generalization beyond synthetic corruptions, we added evaluation on the **DriveAction Benchmark**, whose data are *sourced from real-world mass-production vehicles with strict manual curation*, making it a challenging OOD proxy distinct from DriveRX’s training data (DriveLM).
>
> Using the official evaluation scripts, we compare DriveRX with the base model and several strong VLM agents:
>
> | **Model** | **V-L-A** | **V-A** | **L-A** | **A** |
> | --- | --- | --- | --- | --- |
> | GPT-4o | 88.84 | 84.72 | 86.52 | 81.01 |
> | o3 | 92.19 | 86.61 | 88.66 | 82.23 |
> | Claude 3.7 Sonnet | 86.31 | 80.80 | 82.56 | 80.67 |
> | Align-DS-V (base) | 78.58 | 77.32 | 77.74 | 76.67 |
> | **DriveRX** | **88.09** | **86.56** | **87.17** | **86.18** |
>
> As shown in the table, DriveRX exhibits strong OOD robustness, consistently outperforming the base model across all metrics. It also achieves performance comparable to SOTA closed-source models such as GPT-4o and o3. Notably, on the Action (A) metric, DriveRX even surpasses most SOTA models (86.18 vs. 81.01/80.67). These findings demonstrate that DriveRX generalizes effectively not only to synthetic corruptions but also to unseen, real-world distribution shifts.
>
> ---
>
> ### **Q6 Failure Cases**
>
> We observe that most failure cases arise in scenes where the **Perception stage becomes unreliable due to severe occlusion or missing visual evidence**. Since Perception initializes the entire reasoning chain, inaccuracies at this stage naturally propagate to subsequent Prediction, Planning, and Behavior stages, leading to inconsistencies across reasoning outputs. In addition, in a small number of low-confidence scenarios, DriveRX adopts a **more conservative reasoning strategy**, re-checking ambiguous cues and occasionally generating longer reasoning traces, which slightly increases the inference time. These behaviors occur only under **highly uncertain visual conditions** and do not affect overall performance or latency in normal cases. We will further discuss these limitations in the limitation section of the revised manuscript.
>
> ---
>
> We thank the reviewer once again for the valuable feedback, and we will revise the paper accordingly in the updated PDF.

---

> ### Comment · Reviewer_7UfE · 2025-11-20
> **Follow-up Questions on Q2 and Q3**
>
> Thank the authors for the detailed rebuttal. It addressed most of my concerns. I have a few additional follow-up questions:
>
> Q2: Since “behavior is defined as a discrete abstraction of the ego vehicle’s final driving decision,” what is the relationship between the action in VLA and the behavior here? Should we interpret behavior as essentially representing the action? In other words, is this concept closer to VLB or VLA?
>
> Q3: Have the authors explored a supervised fine-tuning (SFT) approach for training the model? (No need to perform additional experiments, I am simply curious.) I recently read several papers discussing the differences between SFT and RL, and would be interested in any insights or phenomena the authors may have observed during their experiments.
>
> I will update my score once the revised PDF is uploaded.

---

> ### Author Response · Authors · 2025-11-22
>
> Thank you for your follow-up. We are glad to continue the discussion and address Q2 and Q3 below.
>
> ### **Q2 Relationship between Behavior and Action**
>
> Thank you for this insightful follow-up. You are correct that the distinction between "behavior" and "action" is critical in our formulation. In our framework, Behavior represents the high-level semantic decision (expressed in natural language), whereas Action corresponds to the low-level execution (expressed as trajectory waypoints or control signals).
>
> To address your specific questions:
>
> - Relationship between Action and Behavior: Behavior is a VQA task. The output is a natural language description of the ego vehicle's intent (e.g., *"The ego vehicle is steering to the right to avoid the obstacle"*). It serves as the interpretable conclusion of the reasoning chain (Perception → Prediction → Planning). Action: This is the numeric realization of the behavior, explored in our Applications **section (Section 6, DriveRX-Agent/VLA). The output consists of discretized tokens representing physical trajectories or control commands (e.g., throttle, steer).  The Behavior stage acts as a semantic bridge. It summarizes the reasoning process into a decisive intent, which then guides or supervises the generation of precise Action tokens.
> - The Behavior stage in our core AutoDriveRL framework is definitely closer to the concept of VLB (Vision-Language-Behavior). It utilizes the VLM's linguistic capabilities to formulate a decision without directly outputting physical control signals. In contrast, the downstream DriveRX-VLA model (discussed in Section 6) takes this a step further to bridge the gap to VLA (Vision-Language-Action), converting the reasoning-rich representations into executable actions.
>
> ---
>
> ### **Q3 Exploration of SFT and Insights**
>
> Yes, we have explored SFT extensively and provide a quantitative comparison in Appendix K (Table 9). Based on our experiments with DriveRX-SFT (trained on raw annotations) and DriveRX-VLA (distilled from the RL model), we observed distinct behaviors regarding the role of reasoning data:
>
> - **SFT struggles with complex reasoning without explicit thought process:** As shown in Table 9, DriveRX-SFT performs poorly on the high-level Behavior task (scoring 15.34 vs. 62.02 for RL). We attribute this to the "reasoning gap." The original dataset provides only the final ground truth (e.g., "turn left") without the intermediate reasoning chain. SFT models tend to shortcut to the answer based on superficial visual cues, failing to generalize in complex scenarios where structured reasoning (Perception → Prediction → Planning) is required.
> - **RL generates reasoning, while SFT effectively distills it:** While SFT failed on raw data, it succeeded in our Data Distillation experiments (Section 6). The DriveRX-VLA and DriveRX-Agent models, which were SFT-trained on the reasoning traces generated by our RL-optimized DriveRX, achieved competitive performance.
>
> Insights:
>
> - **SFT tends towards memorization over generalization.** Recent research suggests that standard SFT primarily fits the training distribution and often struggles with out-of-distribution robustness [1]. This explains why DriveRX-SFT, trained on static labels, failed to generalize to the complex reasoning required in autonomous driving.
> - **RL incentivizes the emergence of reasoning capabilities.** As demonstrated in recent reasoning-focused architectures [2], RL is essential for incentivizing the exploration of valid Chain-of-Thought (CoT) paths that are not present in raw annotations. This is evidenced by the improved logic observed during our RL training.
> - **Distillation is the optimal role for SFT.** Once high-quality reasoning traces are discovered via RL, SFT becomes a highly efficient mechanism for "distilling" these capabilities into downstream models [2,3]. This validates our two-stage pipeline: using RL to create the reasoning capability (DriveRX) and SFT to transfer it to actionable agents (DriveRX-VLA).
>
> [1] Chu T. et al. Sft memorizes, rl generalizes: A comparative study of foundation model post-training[J]. arXiv preprint arXiv:2501.17161, 2025.
>
> [2] Guo D. et al. Deepseek-r1: Incentivizing reasoning capability in llms via reinforcement learning[J]. arXiv preprint arXiv:2501.12948, 2025.
>
> [3] K. Lu and Thinking Machines Lab, “On-Policy Distillation”, *Thinking Machines Lab: Connectionism*, Oct. 2025. Available: https://thinkingmachines.ai/blog/on-policy-distillation/
>
> ---
>
> We hope our responses address your questions, and we would be happy to discuss further. The revised PDF has been updated; please let us know if anything else needs adjustment. Thank you again for your insightful comments.

---

> ### Author Response · Authors · 2025-11-27
>
> Dear Reviewer **7UfE**,
>
> Hello! Thank you again for your constructive and insightful comments. We have carefully updated our responses accordingly and have also completed the revised PDF. We kindly invite you to take another look and reevaluate our work in light of our clarifications.
>
> If our responses have addressed your concerns, we would greatly appreciate it if you could kindly reconsider the score. If there are any remaining questions or suggestions, we are more than willing to engage in further discussion and make corresponding improvements.
>
> We sincerely appreciate your time and consideration.
>
> Sincerely,
>
> The authors

---

### Author Response · Authors · 2025-11-22

Dear Reviewers:

Hello! We have updated the responses and manuscript to your constructive and insightful comments, and we would like to kindly ask you to take a look at our responses and reevaluate our work based on our clarifications. Please let us know whether our response addresses your concerns or whether there is any further detail we can provide to help address them. We appreciate your time and consideration!

The authors.

---

### Author Response · Authors · 2025-12-02

Dear Area Chair,

We are grateful for your time and effort in handling our paper. As the discussion period concludes, we provide a structured summary below covering our core contributions, the key concerns addressed during the rebuttal, and the current status of reviewer engagement to assist in your final decision.

### **1. Key Contribution:**
We propose AutoDriveRL, a unified reinforcement learning framework that jointly optimizes a vision-language model (DriveRX) across perception, prediction, planning, and behavior using fine-grained task-specific rewards, establishing a robust and interpretable high-level semantic reasoning backbone that significantly outperforms state-of-the-art baselines in complex driving scenarios.

### **2. Summary of Main Concerns & Responses**
**Q1. Motivation and Application**

We thank Reviewers 3U7X and 7UfE for their constructive comments and the opportunity to clarify the motivation and intended application scope of DriveRX.

DriveRX is not designed to serve as a real-time low-level planner or vehicle-control module. Rather, it is built as a cross-task vision–language reasoning backbone, focusing on high-level semantic understanding, intention prediction, and interpretable decision reasoning.

Our results on DriveBench and DriveLM-Hard demonstrate that DriveRX produces stable, coherent, and interpretable reasoning chains even in complex and long-tail scenarios. This validates DriveRX as an effective backbone for high-level semantic reasoning and consistent cross-task decision making. The stage-wise structured outputs also offer practical utility for annotation assistance, hard-case diagnosis, and reasoning-quality inspection.

Furthermore, DriveRX’s high-level reasoning signals can act as teacher supervision or reward feedback for downstream planning and control models. To illustrate this, Section 6 presents two applications—DriveRX-Agent and DriveRX-VLA—showing how “reasoning-enhanced” supervision benefits downstream components.

In summary, this work does not aim to report improvements in real-time low-level driving control. Instead, it establishes a reasoning-centered VLM backbone that unifies semantic perception, prediction, planning, and behavior abstraction, and that can support and enhance downstream autonomous driving modules. While leveraging DriveRX’s structured representations to further improve low-level models is promising, it lies beyond the current scope and is left for future work.

**Q2 More OOD Evaluation**

We clarify that our original submission already includes OOD evaluations using DriveBench. To provide a more complete analysis of model robustness, we now further augment these results with an additional real-world OOD evaluation on DriveAction.

(1) Corruption robustness on DriveBench.
As shown in Table 1, DriveRX is evaluated under the corruption-averaged protocol of DriveBench. In this setting, each image is perturbed by one of 15 diverse synthetic corruption types, including weather-related and sensor-related degradations such as Brightness, Snow, Fog, Lens Obstacle, Water Splash, Camera Crash, Frame Lost, Motion Blur, Zoom Blur, Bit Error, Color Quantization, and H.265 Compression. This setup imposes heterogeneous and substantial input shifts, providing a rigorous robustness test beyond mild or isolated noise.

(2) Real-world OOD robustness on DriveAction.
To further assess generalization beyond synthetic perturbations, we additionally evaluate DriveRX on the DriveAction Benchmark. DriveAction consists of real-world data collected from mass-production vehicles, with strict manual curation to ensure high-quality annotations. As a challenging OOD proxy distinct from DriveRX’s training data (DriveLM), DriveAction allows us to more comprehensively measure the model’s real-world robustness and generalization capability.

Utilizing the official evaluation scripts, we compare DriveRX against the base model and SOTA VLM agents:

| **Model** | **V-L-A** | **V-A** | **L-A** | **A** |
| --- | --- | --- | --- | --- |
| GPT-4o | 88.84 | 84.72 | 86.52 | 81.01 |
| o3 | 92.19 | 86.61 | 88.66 | 82.23 |
| Claude 3.7 Sonnet | 86.31 | 80.80 | 82.56 | 80.67 |
| Align-DS-V | 78.58 | 77.32 | 77.74 | 76.67 |
| **DriveRX** | **88.09** | **86.56** | **87.17** | **86.18** |

The results demonstrate strong OOD robustness. DriveRX significantly outperforms the base model (Align-DS-V) across all metrics on this unseen dataset. Notably, it achieves performance on par with SOTA closed-source models (GPT-4o, o3) and even surpasses most of them on the Action (A) metric (86.18 vs. 81.01/80.67), validating its capability to generalize to complex, real-world driving scenarios.

---

> ### Author Response · Authors · 2025-12-02
>
> **Q3: Clarification on additional ablation studies.**
>
> We would like to clarify that the requested ablation experiments and experimental settings were actually included in the supplementary material. Due to space limitations in the main text, these results were placed in the appendix, which might have made them less visible. Specifically:
>
> - Ablations on the SFT paradigm and RL strategies are provided in Appendix K, analyzing the impact of different training paradigms.
>
> - Stage-wise ablations on the contribution of each training phase are included in Appendix G, showing the incremental gains when adding each component.
>
> - Hyperparameter configurations and related sensitivity analyses are detailed in Appendix O, ensuring reproducibility.
>
> - Additional case studies are presented in Appendix L, offering qualitative insights into model behavior across diverse scenarios.
>
> - Data-filtering ablations are examined in Appendix G, evaluating different filtering strategies to demonstrate their contribution to stability and final performance.
>
> - Rule-based reward ablations are discussed in Appendix J, comparing the effectiveness of rule-based versus hybrid reward mechanisms.
>
> ---
>
> ### **3. Status of Reviewer Engagement**
> We also wish to provide the AC with brief context regarding the review process. We have had highly constructive and meaningful interactions with most reviewers during the rebuttal phase. Reviewers 7UfE, 6G7c, and 3U7X all engaged with us positively and deeply:
> - **Reviewer 7UfE** further discussed the SFT–RL insights with us after reading our rebuttal, expressed that our clarifications addressed their concerns, and indicated willingness to adjust the score after we revise the PDF.
> - **Reviewer 6G7c** focused on detailed experimental aspects and raised multiple technical questions, all of which we carefully addressed. Their feedback was helpful and grounded in the experimental design.
> - **Reviewer 3U7X** provided substantial constructive suggestions. After our rebuttal, we continued deeper discussion, including a practitioner-oriented analysis of DriveRX’s advantages over proprietary API-based systems. Their review meaningfully contributed to improving the paper.
>
> In contrast, we are seriously concerned about the evaluation from **Reviewer Xhmu**, which shows clear signs of **bias and factual misunderstanding**.
>
> Reviewer Xhmu’s primary weakness primary weakness is that DriveRX shows limited improvement on the driving control task. However, driving control performance is not the main target of our method, and we explicitly discuss this in the Limitation section. Penalizing the paper for a non-goal of the method is misaligned with the paper’s actual scope.
>
> More importantly, Reviewer Xhmu claimed that the setup of Table 3 is unclear and that we did not compare against multiple baselines. This is **indeed factually incorrect**: in the original ICLR submission, on page 9, section 6, Table 4, we clearly compare against the baselines the reviewer asserted were missing. This indicates that the reviewer overlooked essential content of the paper, raising concerns about the accuracy and fairness of the evaluation.
>
> Furthermore, while several reviewers mentioned that the motivation could be clearer, none of them considered this severe enough to merit a score as low as 2. This outlier score combined with factual mistakes and the reviewer’s refusal to communicate during the rebuttal—unlike the other reviewers who all engaged actively—raises concerns about fairness and professionalism. Such lack of interaction contradicts the ICLR rebuttal principle, which encourages iterative clarification between authors and reviewers.
>
> Given these issues, we respectfully request that the AC treat Reviewer Xhmu’s evaluation with caution, as the review contains factual inaccuracies, misaligned criticisms, and a disproportionately low score inconsistent with other reviewers’ assessments.
>
> ---
>
> The above constitutes our overall rebuttal. We sincerely thank the AC and all reviewers for their time and constructive comments. We have revised our PDF accordingly and highlighted all changes in red for clarity. We hope that the AC will take these clarifications and improvements into consideration when making the final decision, and we genuinely believe that the strengthened manuscript now more clearly demonstrates the value and contribution of our work.

---

### Meta-Review · Area_Chair_qEj2 · 2026-01-07

**Summary:**

This paper presents DriveRX, a vision-language-action model designed to enhance reasoning and decision-making for autonomous driving. The authors aim to leverage language-based reasoning to handle complex or ambiguous driving cases and show some performance improvements over generalist vision-language models. The model is trained via the AutoDriveRL framework, which decomposes into perception, planning, and behavior tasks with task-specific reward models. The paper also includes an evaluation on a trajectory prediction benchmark to demonstrate the model’s potential for better planning.

Reviewers raised concerns including:
(1) Experimental setup and explanation (and motivation for VLM usage and the resulting model). This includes possible data leakage between DriveBench and DriveRX both based on DriveLM.
(2) Lack of comparison to existing baselines (e.g. UniAD, VAD, DriveVLM, OmniDrive, ORION), with only marginal gains expected.
(3) Lack of temporal information (images or ego state).
(4) Lack of latency discussion.
(5) Gap in SFT and RL performance and ambiguous RL details.
(6) Lack of ablations (e.g. reward composition, GRPO vs PPO, per-task weighting, data filtering).
(7) Training and evaluation both by LLM, without correlating to human labels.
(8) Response of model to distribution shifts is not considered beyond corruption.
(9) Lack of discussion of failure cases emerging due to cross-task reasoning design.

**Reviewer Concerns:**

Authors briefly expanded their description of the experimental setup, and added comparison to baselines UniAD, VAD, and ORION in Table 4. Latency and gaps between SFT and RL are discussed and added during the rebuttal period. The lack of temporal information remains outstanding - while replied in the discussion, the reviewer's point about the importance of this information remains.

Explanations of RL methods are expanded, and ablation studies are clarified by training details presented in the discussion and added to the appendix. Further, explanation that LLM output is validated by three human annotators is added in the discussion period. Failure cases are addressed and added in revised manuscript, and the authors clarify that evaluation already contains examples that are OOD and not just sensor failures.

**Reviewer Scores:**

Xhmu: 2, expected to stay the same due to marginal gains over baselines, excluded from the original manuscript and requested by the reviewer.
3U7X: 6, expected to stay the same based on discussion.
6G7c: 6, expected to stay the same; manuscript is marginally adjusted in accordance with reviewer suggestions.
7ufe: 4, expected to stay the same or increase; this reviewer wanted to re-examine the revised manuscript, but from the discussion, it does appear that suggested changes were taken into account, including discussion of failure cases.

---

### Decision · Program_Chairs · 2026-01-26

Reject